# Engineered alcohol oxidases catalyse transesterification in aqueous media without competing hydrolysis

Bin Wu[1], Yunjian Ma [1,2] ✉, Chenhao Feng[1], Limei Ren[3,4], Chiara Domestici[3], Yutong Wang [3], Thomas Hilberath[3], Ulf Hanefeld [3], Evgeny A. Pidko [5], Frank Hollmann [3] ✉ & Yonghua Wang[1,6] ✉

Transesterification reactions are fundamental transformations in organic chemistry, yet performing them in aqueous media is challenging because of the competing hydrolysis reaction. In this study, we describe a mutant of alcohol oxidase from *Phanerochaete chrysosporium* (*Pc*AOx-VPN) that also exhibits transesterification activity. Moreover, *Pc*AOx-VPN displays no detectable hydrolytic activity, owing to its hydrophobic active site, which effectively excludes water. These characteristics make *Pc*AOx-VPN a promising catalyst for transesterification reactions in aqueous media, a context that is typically compromised by competing hydrolysis.

Transesterification is a cornerstone transformation in organic chemistry, widely exploited to alter the physical and chemical properties of esters and to enable the synthesis of a broad spectrum of compounds, from pharmaceuticals to biodiesel. Conventional approaches typically rely on aprotic solvents, as water is incompatible due to the prevalence of competing hydrolysis reactions. As a result, the overwhelming majority of catalytic transesterifications are performed in non-aqueous media[1,2]. An exception is the acyl-transferase from *Mycobacterium smegmatis* (*Ms*Act)[3], which has attracted considerable attention owing to its preference for alcohol nucleophiles over water, thereby enabling transesterification in aqueous systems[4–10]. Nevertheless, *Ms*Act-mediated reactions remain hampered by hydrolysis as the dominant side reaction[2,4,7,11–18], necessitating stringent kinetic control to mitigate product loss (Fig. 1).

Here, we report an alternative biocatalytic route to transesterification in aqueous media employing an enzyme class not previously associated with this reactivity. Specifically, we identify variants of the flavin-dependent oxidase from *Phanerochaete chrysosporium*[19–21] that catalyse transesterification without concomitant hydrolysis.

## Results and discussion

### *Pc*AOx-VPN exhibits an enlarged substrate scope

Inspired by the foundational work of Fraaije and colleagues utilising *Pc*AOx as oxidation catalyst[20,21], our initial objective was to broaden the substrate scope of *Pc*AOx to accommodate sterically bulkier alcohols than the small, primary alcohols typically processed by the wild-type enzyme. Utilising the crystal structure of *Pc*AOx[20] as a guide, we identified three sites (M59, Q60, and R61) for saturation mutagenesis. Their position at the active site's entrance (Fig. 1), potentially obstructing access for larger substrates, rendered them prime candidates for mutagenesis.

By simultaneously randomising positions 59, 60 and 61, we identified the triple mutant M59**V**-Q60**P**-R61**N** (*Pc*AOx-VPN) to oxidise benzyl alcohol (SI 1.3). The purified enzyme (Supplementary Fig. S2) displayed a bright yellow colour, showcasing the distinctive flavin absorption peaks at 359 nm and 457 nm (Supplementary Fig. S3). Relative to the wild-type enzyme, *Pc*AOx-VPN demonstrated markedly increased activity towards bulkier substrates. For instance, its efficacy in catalysing benzyl alcohol oxidation increased over tenfold compared to the wild-type (0.72 U×mg⁻¹ for wt-*Pc*AOx and 7.7 U×mg⁻¹ for

[1]School of Food Science and Engineering, South China University of Technology, Guangzhou, China. [2]State Key Laboratory of Advanced Papermaking and Paper-based Materials, South China University of Technology, Guangzhou, China. [3]Department of Biotechnology, Delft University of Technology, Delft, The Netherlands. [4]Department of Chemical Engineering, Shijiazhuang University, Hebei, China. [5]Department of Chemical Engineering, Delft University of Technology, Delft, The Netherlands. [6]Guangdong Youmei Institute of Intelligent Bio-manufacturing Co., Ltd, Foshan, Guangdong, China. ✉e-mail: myj605740779@scut.edu.cn; f.hollmann@tudelft.nl; yonghw@scut.edu.cn

*Pc*AOx-VPN, respectively). Simultaneously, its catalytic activity towards methanol diminished from 66.3 U×mg⁻¹ (wt-*Pc*AOx) to 0.16 U×mg⁻¹ (*Pc*AOx-VPN).

To further understand the molecular basis of this change in substrate preference, the crystal structure of *Pc*AOx-VPN was elucidated (SI 2.15). When juxtaposed with the structure of the wild-type enzyme[20], it becomes evident that the M59V substitution plays a pivotal role in broadening the substrate scope of *Pc*AOx-VPN (Fig. 2). In the wild-type enzyme, M59 situates W560 directly atop the flavin moiety, obstructing access for bulkier substrates. Conversely, in *Pc*AOx-VPN, W560 adopts an outward flip, thereby creating additional space above the flavin entity.

## *Pc*AOx-VPN catalyses acyl transfer reactions

With the goal of enhancing the solubility of predominantly hydrophobic substrates, we subsequently explored various deep eutectic solvents, such as a blend of propane-1,2-diol, choline chloride, and $H_2O$ in a 1:1:1 molar ratio. In this milieu, unforeseen additional peaks emerged in the gas chromatograms of the ethyl acetate-extracted reaction mixtures, which, upon GC/MS analysis, were identified as propane-1,2-diol mono- and diacetates (data shown in the source data file). Notably, these products were solely detected in control reactions involving *Pc*AOx-VPN and subsequent extraction with ethyl acetate, leading us to hypothesise a potential promiscuous acyl transferase activity of *Pc*AOx-VPN. To our knowledge, this represents the first example of a flavoprotein catalysing acyl transfer reaction, and prompted us to divert our focus from the alcohol oxidation capabilities of *Pc*AOx-VPN to investigate this newfound activity.

Control reactions in the absence of or using thermally inactivated *Pc*AOx-VPN did not yield detectable amounts of the ester thereby excluding a possible contribution of trace amounts of imidazole (representing possible contaminants from enzyme purification) in the transacylation reaction. Classical serine hydrolase inhibitors, such as phenylmethylsulfonyl fluoride, did not affect the transesterification activity (Supplementary Fig. S10).

Next, we tested several acyl donors. No esterification activity of *Pc*AOx-VPN was detected when the enzyme was incubated with acetic acid and either 2-phenylethanol or benzyl alcohol; even after extended reaction periods ester formation was absent. This lack of activity is likely due to the acyl donor (acetic acid) being predominantly deprotonated and, consequently, unreactive. Activated vinyl esters[22] such as vinyl acetate, isopropenyl acetate, and trifluoroethyl acetate as acyl donors accelerated the esterification reaction of e.g. 2-phenylethanol (Supplementary Fig. S11). Among these, trifluoroethyl acetate clearly outperformed the other acyl donors in terms of product formation. Compared to ethyl acetate, the yield of phenylethyl acetate was ~2.5-fold higher. We attribute this to the higher intrinsic reactivity of trifluoroethyl acetate (kinetic acceleration of the transesterification reaction). But also a more favourable reaction equilibrium likely contributes as with vinyl acetate and isopropenyl acetate, where the byproducts readily tautomerise to the corresponding non-reactive carbonyl compounds, higher yields were observed. In view of avoiding halogenated reagents, we proceeded with vinyl acetate as the acyl donor, which afforded approximately double the initial reaction rates and final product concentrations (Supplementary Fig. S12).

An interesting observation was made for the 'dual-reactivity' substrate benzyl alcohol, which can be both oxidised and transesterified by *Pc*AOx-VPN. The overall reaction rate remained unchanged, whether benzyl alcohol underwent oxidation alone or both oxidation

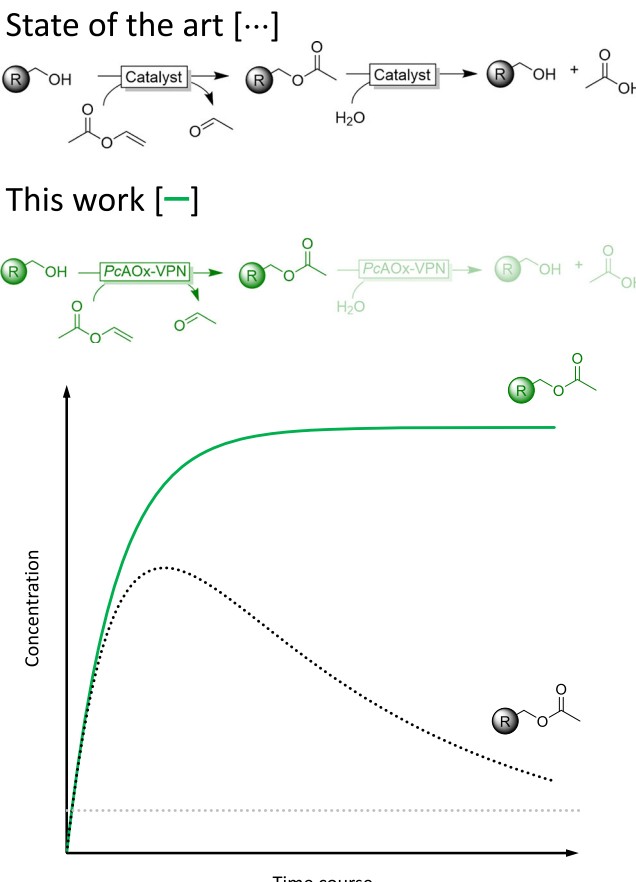

**Fig. 1 | Comparison of the progress of catalytic transesterification reactions in aqueous media.** Established catalysts such as MsAct promote both the desired transesterification reaction and the (undesired) hydrolysis (black, dotted line). In contrast, the PcAOx presented here catalyses exclusively the transesterification (green, solid line). The grey line represents the thermodynamic equilibrium.

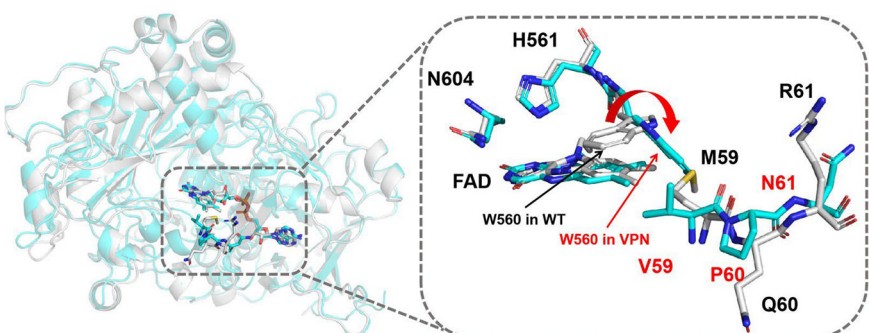

**Fig. 2 | Structural comparison between mutant *Pc*AOx-VPN (cyan, PDB: 9V6K) and wild-type PcAOx (grey, PDB: 6H3G).** The red arrow emphasises the reorientation of W560.

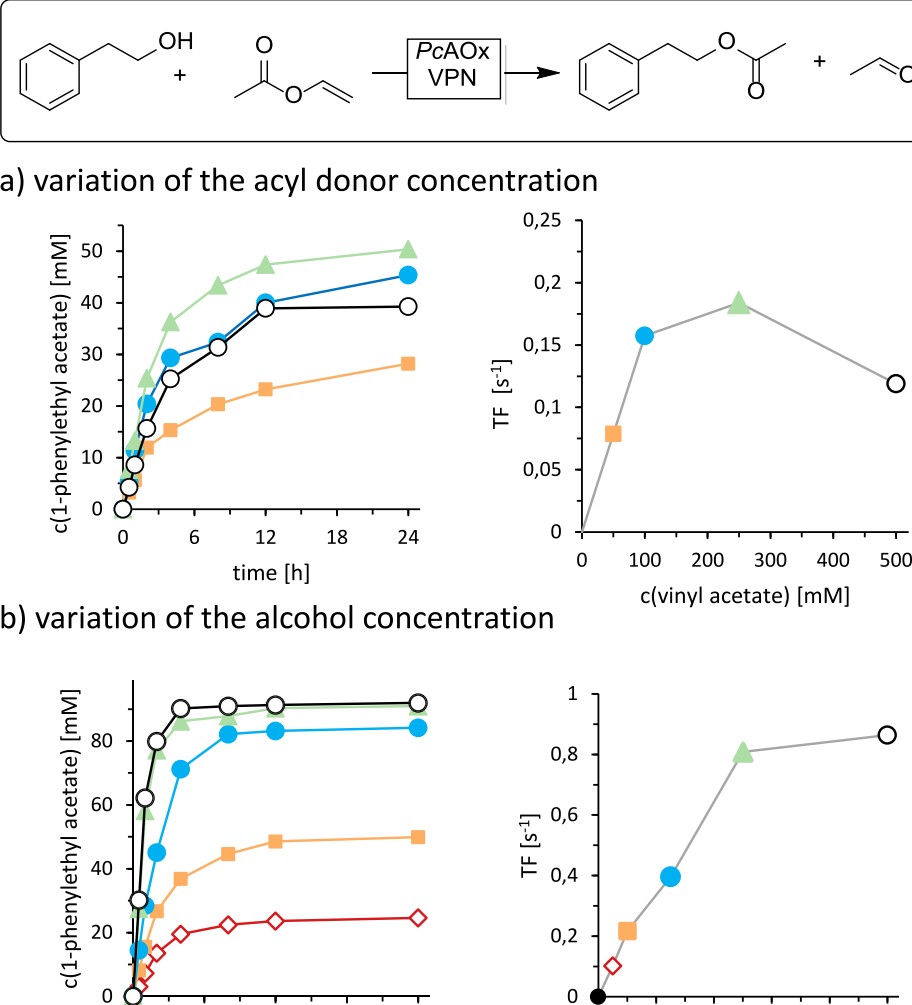

**Fig. 3 | Influence of varying vinyl acetate concentrations on the esterification of 2-phenylethanol.** General conditions: [$Pc$AOx·VPN] = 20 μM, 50 mM NaPi buffer (pH 7.5) containing 5% (v/v) DMSO, 30 °C, 500 rpm. Left: time courses of the ester formation, right: initial rates depending on the varying reagent concentration. **a** Variation of the acyl donor concentration: c(2-phenylethanol) = 50 mM, c(vinyl acetate) = 50–500 mM; **b** Variation of the alcohol concentration: c(vinyl acetate) = 250 mM: c(2-phenylethanol) = 25–500 mM. Experiments were performed as technical duplicates ($N$ = 2), no statistical analysis was performed. The initial rates shown were estimated from the linear portion of the time courses, typically within the first 30–60 min, based on 2–3 data points.

and transesterification simultaneously (Supplementary Fig. S13). This may hint towards substrate binding being the overall rate-limiting step in the catalytic mechanisms of both benzyl alcohol oxidation and benzyl alcohol transesterification. It is also worth noting that the initial transesterification rate was approximately 5 times higher than the oxidation rate.

Next, we investigated the effect of the reagent concentration on the initial rate of the transesterification of 2-phenylethanol with vinyl acetate (Fig. 3). We chose 2-phenylethanol as acyl acceptor as it was only sluggishly oxidised by $Pc$AOx-VPN (Supplementary Fig. S9).

For the alcohol substrate a typical saturation-type dependence of the initial reaction rate on the substrate concentration was observed (Fig. 3b). For the acyl donor, however, substrate inhibition was detected (Fig. 3a). Analogous patterns were observed when employing benzyl alcohol as alcohol substrate (Supplementary Fig. S14). As enzyme inactivation by the reactive acetaldehyde by-product (obtained both, from the enzymatic transesterification reaction and

from vinyl acetate autohydrolysis (~2.5 mM h⁻¹, Supplementary Fig. S17)) may account for the reduced activity and incomplete conversion, we also tested the influence of different isopropenyl acetate concentrations (Supplementary Fig. S15). Again, a pronounced substrate inhibition was observed. This may be the result of substrate inhibition originating from an ordered bi-bi ping-pong mechanism (Supplementary Fig. S16), in which the alcohol binds first, followed by the acyl donor. However, more detailed kinetic studies are required to confirm this hypothesis.

Notably, ester yields (based on the limiting substrate) generally exceeded 80%, except in cases where both 2-phenylethanol and vinyl acetate concentrations were high (Fig. 3b). Under these conditions, no more than 90 mM of the corresponding 2-phenethyl acetate was obtained. A fully satisfactory explanation is currently lacking; however, product inhibition and reduced enzyme stability in the presence of high concentrations of organic compounds may account for this observation. Further studies will address this issue.

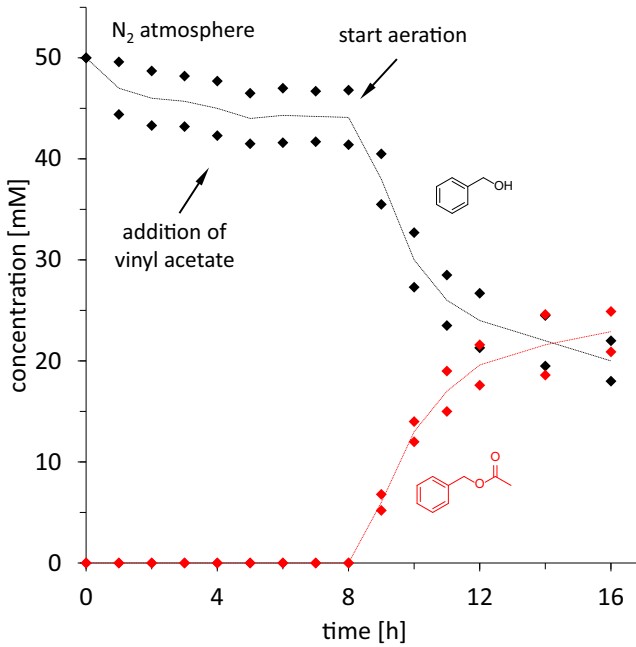

**Fig. 4 | Influence of the FAD oxidation state on the PcAOx-VPN transesterification activity.** Reaction conditions: [*Pc*AOx-VPN] = 50 μM, [benzyl alcohol] = [vinyl acetate] = 50 mM in 50 mM NaPi buffer (pH 7.5) containing 5% v/v DMSO, *T* = 30 °C, 500 rpm. Shortly after addition of the substrates under anaerobic conditions, the reaction mixture decolorised indicating the reduction of the flavin prosthetic group. After 8 h the reaction mixture was exposed to ambient air resulting in an immediate re-formation of the yellow colour typical for oxidised flavins. Experiments were performed as technical duplicates (*N* = 2), no statistical analysis was performed. For reasons of clarity, benzaldehyde is not shown.

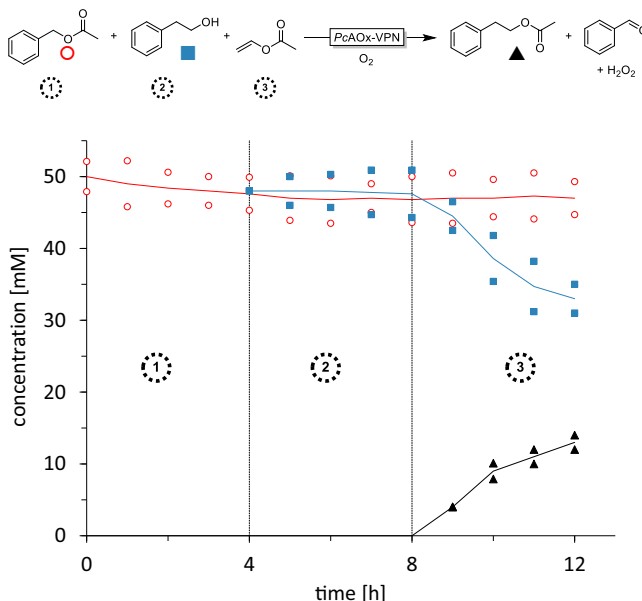

**Fig. 5 | Investigation of the hydrolytic activity of *Pc*AOx-VPN.** Reaction conditions: [*Pc*AOx-VPN] = 20 μM, 50 mM phosphate buffer (pH 7.5) containing 10% v/v DMSO, T = 30 °C, 500 rpm. Phase 1: at *t* = 0 h 50 mM benzyl acetate was added; phase 2: at *t* = 4 h 50 mM 2-phenylethanol was added; phase 3: at *t* = 8 h 50 mM vinyl acetate was added. Experiments were performed as technical duplicates (*N* = 2), no statistical analysis was performed.

## Oxidised *Pc*AOx-VPN catalyses the transesterification reaction

Given that *Pc*AOx-VPN is a flavoprotein, the oxidation state of the enzyme-bound FAD group could impact its transesterification activity. To investigate this, we devised an experiment investigating the transacylase activity of reduced and oxidised *Pc*AOx-VPN (Fig. 4). Under anaerobic reaction conditions and using oxidisable benzyl alcohol as acyl acceptor no benzyl acetate formation was observed in the presence of vinyl acetate (acyl donor) over 4 h of incubation time. Upon introduction of O₂ to the reaction mixture, a rapid reoxidation of the flavin cofactor was observed (also visible by the re-formation of the characteristic yellow colour) together with a commencing formation of benzyl acetate.

## *Pc*AOx-VPN does not catalyse ester hydrolysis

Another unexpected observation was the absence of ester hydrolysis. The concentration of benzyl acetate in the presence of *Pc*AOx-VPN remained essentially unchanged, even after prolonged incubation (Fig. 5), and was unaffected by the addition of 2-phenylethanol. Transesterification occurred only upon addition of vinyl acetate, yielding 2-phenylethyl acetate. Throughout this experiment, neither a significant decrease in benzyl acetate concentration nor the formation of benzyl alcohol or benzaldehyde was detected, indicating that hydrolysis of benzyl acetate did not take place. Further control experiments confirmed these results (Supplementary Fig. S18). It is also noteworthy that 2-phenylethyl acetate was apparently not formed via transesterification of benzyl acetate. This suggests that the active site of *Pc*AOx-VPN may not be sufficiently spacious to simultaneously accommodate both bulky substrates, benzyl acetate and 2-phenylethanol.

The absence of hydrolytic activity is consistent with Fraaije's observation that the enzyme's active site is effectively impermeable to solvent (i.e. water) molecules[20]. The active site is lined predominantly by aromatic residues (F101, Y407, F422, W560 and H561), together with the similarly hydrophobic M103, which together account for its pronounced hydrophobic character. In the crystal structure of *Pc*AOx-VPN, no water molecules are observed within 8.2 Å of the flavin cofactor. Analysis with CAVER Analyst[23] (Supplementary Fig. S7) likewise confirms the hydrophobic nature of both the active site and the access channel.

## Substrate scope of *Pc*AOx-VPN

Next, we performed a preliminary evaluation of the substrate scope of *Pc*AOx-VPN (Fig. 6). As mentioned above, several acyl donors such as ethyl acetate, vinyl acetate, isopropenyl acetate, and trifluoroethyl acetate were accepted by *Pc*AOx-VPN. Also benzyl acetate and 2-phenylethyl acetate were accepted as acyl donors but only if combined with small alcohols (Fig. 6, **1a–4a** vs. Fig. 5). The somewhat larger 2-phenylethyl acetate consistently gave lower yields compared to benzyl acetate. In direct comparison (Fig. 6, **4a**), vinyl acetate clearly outperformed the latter two acyl donors. Amongst the aliphatic alcohols tested, primary alcohols (**1–8**) were preferred over the 2-alkanols of the same chain length (**17–21**). Approx. 2-fold more **4a** was formed than **17a** whereas the yield of **8a** was almost 5 times higher than of **21a**. An optimal chain length of 4–5 was observed for aliphatic alcohols. In the case of α,ω-diols, both mono- and diacetylation were observed (as with 1,5-pentanediol, **9**). For 1,8-octanediol (**10**), only the monoacetylated product (**10a**) was detected, which is presumably due to its lower reactivity.

We also performed a set of experiments utilising methyl- or nitro-substituted benzyl alcohols (**11–13** and **14–16**, respectively). With those starting materials, both transesterification and oxidation was observed. With the exceptions of **12** and **13**, the ratio of transesterification product to oxidation product was ~2:1. Electron-donating

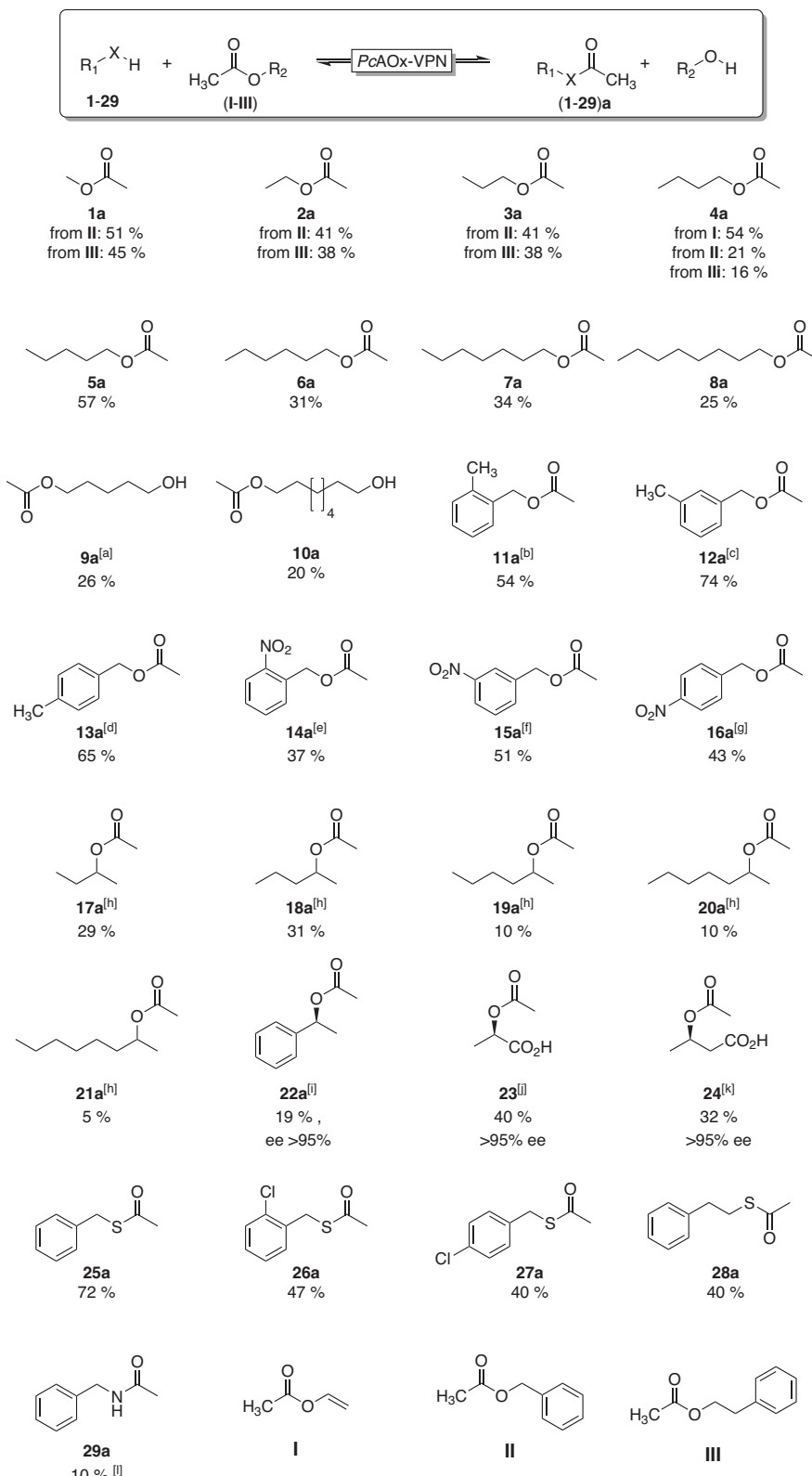

**Fig. 6 | Preliminary substrate scope of *Pc*AOx-VPN.** General conditions: [*Pc*AOx-VPN] = 20 µM, [substrate] = 50 mM in 50 mM KPi buffer (pH 7.5) containing DMSO (5%, v/v) as cosolvent. *T* = 30 °C, 500 rpm, [vinyl acetate] = 250 mM. Experiments were performed as technical duplicates (*N* = 2). The values given represent GC-yields estimated via peak areas. Product formation of compound **1**–**3** was performed indirectly based on the formation of benzyl alcohol and benzaldehyde, respectively. [a] Diester: 12%; [b] aldehyde: 29%; [c] aldehyde: 23%; [d] aldehyde: 19%; [e] aldehyde: 16%; [f] aldehyde: 25%; [g] aldehyde: 21%; [h] starting from racemate, ee was not determined; [i] starting from racemate, E < 48; [j] starting from racemate, E < 75; [k] starting from racemate, E < 61; [l] background subtracted.

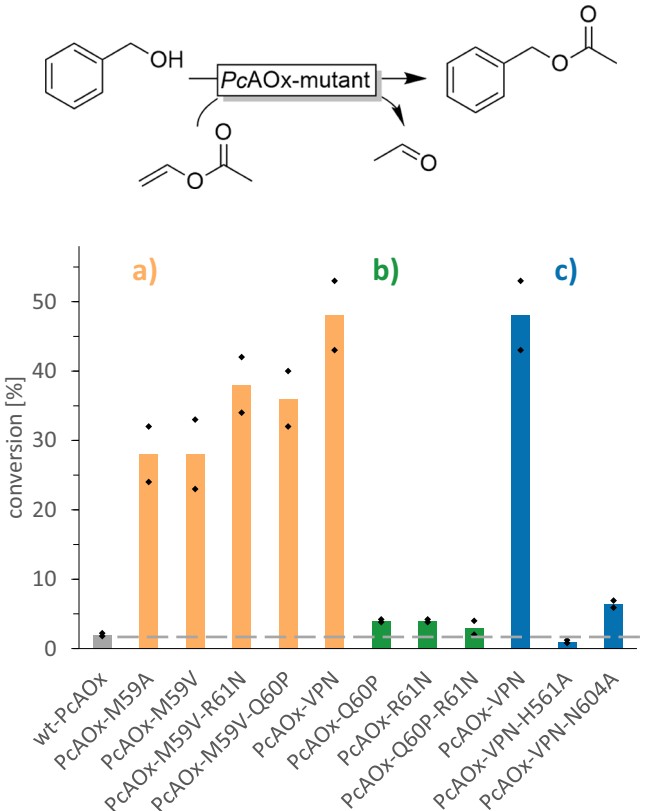

**Fig. 7 | Transesterification activity of wt-PcAOx and some mutants. a** Mutants involving M59A or M59V, **b** 'partial' VPN mutants containing unaltered M59, **c** PcAOX-VPN compared to its H561A and N601A mutants. Reaction conditions: [PcAOx] = 20 μM, [benzyl alcohol] = 50 mM, [vinyl acetate] = 250 mM, in 50 mM NaPi buffer (pH 7.5) containing 5% v/v DMSO, $T$ = 30 °C, 500 rpm, $t$ = 12 h. Experiments were performed as technical duplicates ($N$ = 2), no statistical analysis was performed.

methyl substituents enabled higher reaction rates than electron-withdrawing nitro-substituents for both, the transesterification reaction as well as the concomitant benzylic oxidation reaction. Interestingly, meta-substitution gave the highest activities, which may be attributable to steric effects.

Consistent with findings by Fraaije and co-workers[21], an (S)-preference was observed when testing both individual pure enantiomers and the racemate of 1-phenylethanol. In both cases, no significant conversion of (R)-1-phenylethanol was observed, indicating that the transesterification activity is also highly enantioselective (Supplementary Fig. S20). Likewise racemic lactic acid (Supplementary Fig. S21) as well as 3-hydroxy butyric acid (Supplementary Fig. S22) underwent a kinetic resolution when exposed to PcAOx-VPN in the presence of vinyl acetate. $E$-values ranging from 48 to 75 were estimated.

Docking of the four stereoisomers of the putative tetrahedral intermediates leading to 1-phenylethyl acetate formation into the structure of PcAOx-VPN showed that the intermediates derived from (S)-1-phenylethanol displayed markedly more favourable docking scores than those from the (R)-enantiomer (Supplementary Fig. S8).

We also tested some alternative to alcohols as nucleophiles in the PcAOx-VPN-catalysed acyl transfer. Benzylamine for example, served as nucleophile (Fig. 6, Supplementary Fig. 29a). Due to the considerable extent of spontaneous amidation, a distinct rate acceleration was only observed during the first 30 min of the reaction (Supplementary Fig. S23). Thiols also gave the desired thioester products (Fig. 6, Supplementary Figs. 25a–28a). In this case, however, also a thiol oxidase

activity was observed (as judged by GC/MS data, Supplementary Table S4, entry 23).

To evaluate the preparative scope of the proposed PcAOx-VPN-catalysed transesterification, we performed the acetylation of 2-phenylethanol with vinyl acetate on a 25 mmol scale (Supplementary Section 2.12 in the Supplementary Information). Within 12 h approximately the 2-phenethyl acetate yield was 82%, the isolated yield (2.57 g) was 62% (Supplementary Fig. S24).

In summary, PcAOx-VPN readily accepts various activated acetyl donors and exhibits clear preferences among aliphatic and aromatic alcohols, displaying (S)-enantioselectivity. Its reduced activity with bulkier substrates suggests that PcAOx-VPN is particularly well suited to the biocatalytic synthesis of small to medium-sized esters, whereas conversions of larger substrates may be less efficient and could benefit from further enzyme engineering.

## M59A/V, H561 and N604 are pivotal for activity

To determine which of the newly introduced mutations contributed to transesterification activity, we individually introduced mutations at positions M59, Q60, and R61 (Fig. 6). Only the variants containing M59A or M59V showed a marked increase in catalytic activity compared to the wild-type parent enzyme (Fig. 7), thereby supporting our structural hypotheses (Fig. 2). The individual or combined introduction of Q60P and R61N into PcAOx did not enhance transesterification activity on their own (Fig. 7b). However, when combined with M59A or M59V, these mutations further increased the enzyme's catalytic activity (Fig. 7a).

In addition, we mutated H516 and N604, as these residues have previously been reported as strictly conserved general base/acid catalysts in the oxidation mechanism of wild-type PcAOx (Fig. 6c)[20,24]. These two residues also proved to be crucial for activity in the PcAOx-VPN-catalysed transesterification reaction.

In their natural role as catalytic dyad activating the substrate via alcohol deprotonation His561 acts as general base while Asn604 delocalises the forming positive charge by accepting a hydrogen bond from the pre-existing $N^\varepsilon$–H of His561 (Fig. 8a)[20,24]. In the transesterification reaction, this increases the nucleophilicity of the alcohol for the nucleophilic attack of the ester substrate. The resulting anionic tetrahedral intermediate is likely to be stabilised via electrostatic interaction with the His561/Asn604 cation as well as through H-bonding with Asn604 (Fig. 8b). In subtilisin, an analogous hydrogen-bonding interaction is responsible for an acceleration of ~200–300-fold[25]. Molecular docking of the putative tetrahedral intermediate arising from the nucleophilic attack of (S)-1-phenylethanol on vinyl acetate revealed highly favourable hydrogen-bonding interactions, with distances of 1.8 and 2.8 Å between the oxyanion and the $N^\varepsilon$–H of His561 and the terminal $NH_2$ of Asn604, respectively.

The role of the oxidised flavin cofactor in the proposed catalytic transesterification mechanism remains uncertain. In flavin-dependent HNLs, where the oxidised flavin is likewise required for the non-redox nucleophilic attack of cyanide on a carbonyl group, it has been suggested that FAD is not directly involved in catalysis. Instead, the observed enzyme inactivation upon cofactor reduction is thought to result primarily from a reversal of the active-site electrostatic potential[26]. Nevertheless, conformational changes associated with the bent, reduced flavin may also play a role.

## The VPN motif can be generalised

To examine whether this transacylation capability was unique to PcAOx-VPN, we investigated two other AOxs from Polyporus arcularius (ParAOx) and Aspergillus flavus (AflAOx), which share sequence similarity with PcAOx (ParAOx: 87% identity; AflAOx: 53% identity). The AlphaFoldIII[27]-predicted structures of ParAOx and AflAOx were found to be almost superimposable with the crystal structure of PcAOx-VPN (Supplementary Fig. S6).

**Fig. 8 | Proposal for the mechanistic implementation of His561 and Asn604. a** In the alcohol oxidation, the His/Asn dyad has been proposed to facilitate the hydride abstraction from the alcohol C-H bond. **b** We propose a similar mechanism, in which deprotonation facilitates the nucleophilic attack of the alcohol on the ester carbonyl group, while the resulting transition state is stabilised through hydrogen bonding and electrostatic interactions. Shown here is the molecular docking of the putative tetrahedral intermediate formed upon the attack of (S)-1-phenylethanol on vinyl acetate.

*Par*AOx and *Afl*AOx were recombinantly expressed in *E. coli* BL21(DE3), yielding soluble enzymes with expression levels of 678 and 179 mg × L$^{-1}$ culture medium for *Par*AOx and *Afl*AOx, respectively. Their specific activities for methanol oxidation were 3.6 and 1.1 U × mg$^{-1}$, respectively. Similar to the wild-type *Pc*AOx, native *Par*AOx and *Afl*AOx displayed only minimal activity towards benzyl alcohol as well as in the transesterification reaction. Consequently, mutants of *Par*AOx and *Afl*AOx, mirroring the mutations found in *Pc*AOx-VPN, were engineered. Specifically, the mutants *Par*AOx-M59V-Q60P-R61N and *Afl*AOx-M58V-K59P-L60N (*Par*AOx-VPN and *Afl*AOx-VPN, respectively) were created. Both *Par*AOx-VPN and *Afl*AOx-VPN demonstrated transesterification activities comparable to that of *Pc*AOx-VPN, and, notably, the active-site histidine was again pivotal in the transesterification reaction (Fig. 9). We therefore hypothesise that the here reported transesterification activity of a flavin-containing enzyme is rather a general phenomenon than limited to *Pc*AOx.

In this study, we unveil a yet unknown transesterification activity of flavin-dependent oxidases, marking a significant departure from their known functions. To our knowledge, this 'promiscuous activity' represents a facet of flavoenzyme catalysis not previously documented. In hindsight, this activity is not completely unexpected. The mechanism of FAD-containing alcohol oxidases comprises the activation of the alcohol OH group to facilitate hydride abstraction. This partially deprotonated OH group, however also exhibits higher nucleophilicity thereby preparing it for nucleophilic reactions such as the attack at a ester functionality.

The intrinsic hydrophobicity of the active site in *Pc*AOx plays a pivotal role in kinetically restricting water nucleophile access, effectively minimising unwanted hydrolysis reaction. This unique characteristic facilitates the execution of transesterifications in aqueous environments, eliminating the necessity for kinetic reaction control. This discovery not only broadens our understanding of flavin-dependent oxidases but also opens new avenues for the application of these enzymes in ester synthesis.

## Methods

### Enzyme preparation
Recombinant cells expressing *Pc*AOx were grown in LB medium with kanamycin (50 μg × mL$^{-1}$) at 37 °C to OD$_{600}$ ≈ 0.8 and used (10 mL into 500 mL) to inoculate TB medium (1% glucose, 50 μg × mL$^{-1}$ kanamycin). At OD$_{600}$ 0.8–0.9, cultures were shifted to 20 °C, induced with 0.5 mM IPTG for 20 h, harvested by centrifugation, washed, and stored at −80 °C. Frozen cells (10% w/v) were resuspended in lysis buffer (50 mM potassium phosphate, pH 7.8, 400 mM NaCl, 100 mM KCl, 40 mM imidazole, 100 μM FAD), lysed by sonication, and clarified by centrifugation. The supernatant was applied to Ni$^{2+}$-NTA, *Pc*AOx was eluted with a 40–500 mM imidazole gradient, desalted into 50 mM potassium phosphate (pH 7.5), analysed by SDS–PAGE, and stored at −80 °C.

### Transesterification reactions
Analytical-scale *Pc*AOx-VPN–catalysed transesterifications were carried out in 50 mM sodium phosphate buffer (pH 7.5) containing 5–10%

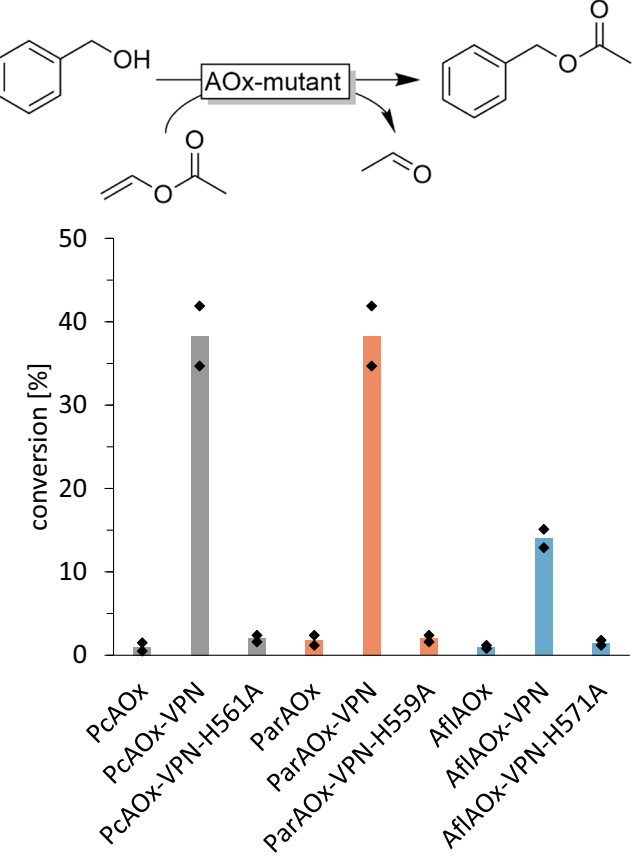

**Fig. 9 | Transesterification activity of several AOx-mutants.** Reaction conditions: [enzyme] = 20 μM, [benzyl alcohol] = 50 mM, [vinyl acetate] = 250 mM, in 50 mM NaPi buffer (pH 7.5) containing 5% v/v DMSO, T = 30 °C, 500 rpm, t = 12 h. Experiments were performed as technical duplicates (N = 2)), no statistical analysis was performed.

(v/v) DMSO at 30 °C and 500 rpm. *Pc*AOx-VPN was used at 20–50 μM, and substrate concentrations were as indicated in the main text, added from DMSO stock solutions (50 μL per reaction). Buffer, substrate and cosolvent were premixed, and reactions were initiated by addition of the enzyme. Unless stated otherwise, reactions were performed under ambient atmosphere and monitored by time-course analysis.

For preparative-scale synthesis, *Pc*AOx-VPN (20 μM), 2-phenylethanol (50 mM) and vinyl acetate (50 mM) were incubated in 500 mL sodium phosphate buffer (50 mM, pH 7.5, 5% v/v DMSO) at 30 °C and 300 rpm for 12 h in a 1 L round-bottom flask under ambient atmosphere. The reaction mixture was filtered and extracted twice with ethyl acetate (250 mL each). Combined organic phases were dried over anhydrous $Na_2SO_4$ and concentrated by rotary evaporation at 40 °C. The crude product was purified by short-path distillation at <1 Pa, sequentially heating to 40 °C (removal of residual solvents), 80 °C (removal of 2-phenylethanol) and finally 120 °C to collect phenethyl acetate. From a theoretical yield of 3.38 g, 2.57 g of purified phenethyl acetate was obtained.

### Reporting summary

Further information on research design is available in the Nature Portfolio Reporting Summary linked to this article.

## Data availability

All data are available in the main text or the supplementary information file, and from the corresponding author(s) upon request. Primary data such as chromatograms are available from Y.M. Crystallographic data has been deposited (9V6K [https://www.rcsb.org/structure/unreleased/9V6K]). The accession numbers for the two newly identified AOx are: *Par*AOx: TFK81242.1 [https://www.ncbi.nlm.nih.gov/protein/TFK81242.1/]. *Afl*AOx: XP_041142246.1 [https://www.ncbi.nlm.nih.gov/protein/XP_041142246.1/]. Source data are provided with this paper.

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

## Acknowledgements
This work was supported by the Program of Natural Science Foundation of China (32402025, B.W.) and the European Union (ERC, PeroxyZyme, No 101054658, F.H.). Views and opinions expressed are however those of the authors only and do not necessarily reflect those of the European Union or the European Research Council. Neither the European Union nor the granting authority can be held responsible for them.

## Author contributions
B.W. designed the study and performed experiments; C.F. was responsible for the crystallographic study; L.R. and E.A.P. performed in silico experiments; C.D, Y.W. and T.H. performed experiments; U.H. was involved in data interpretation and manuscript formulation; F.H. and Y.M. designed experiments and analysed and interpreted experimental data, Y.W. directed the whole project. All authors contributed to the manuscript preparation.

## Competing interests
The authors declare that they have no known competing financial interests or personal relationship that could have appeared to influence the work reported in this paper. Guangdong Youmei Institute of Intelligent Bio-manufacturing Co., Ltd provided some experimental instruments for testing during the whole experiment process. Guangdong Youmei Institute of Intelligent Bio-manufacturing Co., Ltd declares no commercial interest in the results reported in this study.
