## [Transparent Peer Review file · Nature Communications]

Engineered alcohol oxidases catalyse transesterification in aqueous media without competing hydrolysis

Corresponding Author: Professor Frank Hollmann

Version 0:

Reviewer comments:

Reviewer #1

(Remarks to the Author)

The submitted manuscript, entitled "Irreversible transesterification reactions in aqueous media" (NCOMMS-24-71991-T), authored by F. Hollmann, Y. Wang, et al., presents the discovery of a novel engineered variant of alcohol oxidase from *Phanerochaete chrysosporium* (PcAOx-VPN) exhibiting unexpected ('promiscuous') quasi-irreversible transesterification activity in water, without reversed hydrolytic action. The reported biocatalyst is the first known flavin-containing enzyme capable of catalyzing a reaction beyond its native catalytic repertoire. As such, it has the potential to serve as a superior alternative to existing variants of *Mycobacterium smegmatis* acyltransferase (MsAcT).

I find this paper highly interesting and valuable to the broader scientific community, as the reported research presents transformative results that significantly advance the field of bioorganic chemistry. Moreover, it has the potential to open new avenues for future synthetic endeavors, particularly in the transesterification of alcohols. In this context, the study exemplifies the critical need for discovering novel biocatalysts with unnatural features that enable challenging transformations—often unattainable through classical chemical methodologies due to strongly unfavorable thermodynamics. A key strength of this paper is the successful development of the efficient triple mutant M59V-Q60P-R61N (PcAOx-VPN) through saturation mutagenesis. Notably, this variant demonstrated the ability to catalyze the desired transesterification of selected model primary alcohols with bulkier aryl substituents, such as benzyl alcohol and 2-phenylethanol. However, while the newfound activity of the M59V-Q60P-R61N variant is highly promising, there are notable limitations in the tested substrate scope. The study suggests that the method is restricted to a narrow range of substrates, predominantly specific primary alcohols. Interestingly, PcAOx-VPN also catalyzes the efficient alcoholysis of benzyl esters using various aliphatic alcohols (MeOH, EtOH, n-PrOH, and n-BuOH), which could serve as a promising alternative to hydrolytic processes used in applications such as biodiesel production.

The experimental and analytical methods were rigorously designed and executed to meet high standards, ensuring reliability and reproducibility. A thorough analysis of the "Supporting Information" file confirms that the results are well-supported by appropriately designed controls, detailed analyses, and robust statistical evaluation, with no significant flaws or oversights. Technically, the reported data are well-documented, and the paper is written in a clear, concise, and accessible manner. Furthermore, all included figures, tables, and supplementary materials are of high quality, well-organized, and effectively communicate the results.

Given its exceptional quality, novelty, and significant impact, I recommend that this manuscript be considered for publication in Nature Communications. However, I suggest that the following issues be carefully addressed by the authors prior to publication:

Minor points:

(1) In general, the title of the submitted manuscript is slightly misleading, and more suitable for review-type article rather than a communication in Nature journal. In general, it suggests as it report on this topic in a broader context. In my opinion, the title should relate more to discovered variant of alcohol oxidase from *Phanerochaete chrysosporium* (PcAOx-VPN) and its promiscuous catalytical activity in terms of transesterification in water.

(2) In the 'Introduction' section authors stated that: "(...) Serendipitously, we discovered transesterification activity in variants of the alcohol oxidase from *Phanerochaete chrysosporium* (PcAOx)11-13 (...)". After careful reading of all three cited articles, I cannot confirm that any of the authors from the list of the submitted paper participated in those discoveries. Therefore, please rephrase this sentence as it suggests that the authors took part in those studies. As I understand correctly, Prof. Marco W. Fraaije (who is one of the authors of the papers cited in Refs. 11–13) is affiliated with the University of Groningen (NL) and not with TU Delft (NL).

(3) In the 'Introduction' section, there is an insufficient number of references concerning the application of *Mycobacterium smegmatis* acyltransferase (MsAcT). I strongly recommend adding some key citations that highlight the potential of this enzyme in synthetic applications, such as: (i) P. Borowiecki et al. *Angew. Chem. Int. Ed.* 2025, 64, e202420133; (ii) U. T. Bornscheuer et al. *ACS Catal.* 2021, 11, 14906–14915; (iii) U. T. Bornscheuer et al. *Angew. Chem. Int. Ed.* 2020, 59, 11607–11612; (viii) W. Kroutil and F. Himo et al. *ACS Catal.* 2018, 8, 10698–10706; (ix) U. Hanefeld et al. *Adv. Synth. Catal.* 2017, 360, 242–249, etc.

(4) Please refine the following sentences: "(...) The incubation of for example 2-phenylethanol acetate with PcAOx-VPN (...)". The chemical name of "2-phenylethanol acetate" is faulty in this case. It should be "2-phenylethyl acetate".

(5) Out of curiosity, I would like to ask the authors whether they have tested other quasi-irreversible acyl donors (e.g., fatty acid-like vinyl esters, isopropenyl acetate, acetic anhydride, 4-nitrophenyl acetate, 4-chlorophenyl acetate, 2,2,2-trifluoroethyl acetate, etc.).

(6) The authors stated that: "(...) For the acyl donor vinyl acetate, however, substrate inhibition was detected (...)" and then: "(...) We interpret these observations as a result of an ordered, sequential reaction mechanism (...)". Maybe the drop in the catalytic activity is due to the in situ formation of acetaldehyde. As this might be true, once again, I suggest testing other acyl donors which does not generate acetaldehyde byproduct.

(7) The authors present the results of biotransformations using concentrations of the product obtained. I think it would be better to show a conversion (% conv.).

Major points:

(1) This study is significantly limited by its narrow substrate scope, as it primarily focuses on ortho-, meta-, and para-substituted benzyl alcohols. In particular, the absence of racemic secondary alcohols is a major drawback, as their inclusion could provide valuable insights into the catalytic capabilities of the engineered variant, especially regarding enantioselectivity. To address this limitation, I strongly recommend revising the manuscript by applying the PcAOx-VPN variant to the kinetic resolution of representative racemic secondary alcohols. These should include benzylic alcohols, aliphatic alcohols, "small-bulky" aliphatic-aryl alcohols, hydroxy esters, heteroaromatic (halo)methyl alcohols, and "bulky-bulky" aryl-aryl alcohols. Furthermore, all reactions and the resulting optically active products should be fully characterized in terms of their stereochemical outcomes. This should include an assessment of the enzyme's stereopreference, the enantiomeric excess (% ee) of the isolated kinetic resolution (KR) products, and the enantioselectivity (E-value).

(2) I also strongly encourage the authors to test their enzyme on substrates containing more than a single hydroxyl group. Using exemplary diols, triols (such as simple glycerol), and other benchmark multi-hydroxyl compounds could help demonstrate the regioselectivity of the investigated biocatalyst. Additionally, the inclusion of at least one amino alcohol as a substrate would be valuable in assessing the chemoselectivity of the discovered biocatalyst.

(3) The authors stated that the unusual ability of PcAOx-VPN to catalyze transesterification in water without hydrolysis is attributed to the hydrophobic architecture of its active site. To support this hypothesis regarding the catalytic mechanism of action, I strongly recommend visualizing the tunnels and channels in the molecular structure of PcAOx-VPN that lead to the active site entrance. This can be achieved using software such as CAVER Analyst 2.0, developed by researchers at the Loschmidt Laboratories and the National Centre for Biomolecular Research (NCBR) at Masaryk University. Such an analysis could provide deeper insights into the structural factors and molecular basis that determine the rate-limiting step, particularly concerning diffusion effects. Additionally, it would enhance the rationalization of the observed substrate scope.

(4) Since *Nature Communications* values research that bridges disciplines and has applications across multiple fields, I recommend that the authors enhance their paper to align with these standards, ensuring that their work resonates with a broader scientific audience beyond specialists in biocatalysis. To achieve this, I suggest highlighting potential applications of the transesterification methodology in a broader context. For instance, demonstrating its extension toward the synthesis of exemplary small-molecule pharmaceuticals, bioactive compounds, or specialty chemicals would significantly strengthen its relevance. Additionally, this would help justify the selection of model alcohols. Without such examples, the impact of this methodology may be perceived as limited to a narrowly defined set of substrates, potentially restricting its broader applicability and diminishing its potential influence on future discoveries in modern bioorganic chemistry.

(5) In the "Supporting Information" file, I recommend compiling all relevant details regarding the recombinantly produced enzymes into a single table. This table should include information on the vector and expression host used, plasmid number, plasmid vector, and the inducer used for protein expression.

(6) According to the Notice for Authors set by prestigious journals: "(...) For synthetic methods, a selection of substrates illustrative of the scope of the reaction should be made, and the isolated yields of the corresponding purified products should

be reported. Product yields determined by HPLC or GC are considered indicative of a preliminary catalytic study (...)." Therefore, to better illustrate the applicability of the method, I strongly recommend including at least one reaction performed on a gram scale.

Reviewer #2

(Remarks to the Author)

This manuscript describes a protein engineering work on alcohol oxidase from *Phanerochaete chrysosporium* (PcAOx) which led to the identification of a triple mutant that is able to perform a transesterification reaction avoiding undesired hydrolysis occurring with other enzymes. PcAOx can be expressed in *E. coli* and already proved to be prone to rational modification of substrate specificity from alcohols to glycerol, representing an interesting biocatalytic tool. This work is relevant and could have interesting implications, nevertheless I have some issues:

- crystal structure of PcAOx-VPN: a figure showing the electron density of the active site (for example, the map of the FAD cofactor and that of the mutated residues) must be included. Moreover, Table S2 is reporting an R-merge of 27%, 269% in the highest resolution shell: this is quite high whereas the authors declare a CC1/2 of 0.998 (0.720 in the highest resolution shell). How do you explain this? Why the R-merge/R-sym was not reported during PDB deposition as gathered from the table at page 8 of the PDB validation report?

- lines 76-77: Figures S12 and S13 are not related to crystal structures.

- I would suggest to include the term alcohol oxidase in the title because the serendipitous discovery of an alcohol oxidase performing a transesterification reaction is an important aspect of this work.

- Fig. S3: the spectrum shows that the 359 nm is (besides the gap in Abs measurement probably due to reading problems) much higher than the peak at 457 nm, which is unusual in flavoproteins. Although the spectrum is not extended to the UV range, the Abs baseline seems to increase from 700 nm backwards, which is typical of turbidity. Indeed, the protein sample in the picture looks a bit turbid. Can authors comment on this aspect?

- Supplementary page S4: authors should specify what they mean with the primer portions highlighted in yellow. This appears also in other sections of the Supplementary text. One can guess what they mean, but it should be explicitly specified.

Reviewer #3

(Remarks to the Author)

The manuscript submitted by Hollmann, Wang, and co-workers reports the (serendipitous) discovery that a triple variant of alcohol oxidase from *Phanerochaete chrysosporium* (PcAOx-VPN) catalyses transesterification reactions of several simple primary and secondary alcohols with vinyl acetate (and a few additional acetyl esters) in aqueous medium, notably without a detectable hydrolysis side activity. The authors explore the scope of this transformation, show that it can be generalised to other alcohol oxidases, and make some proposals regarding its catalytic mechanism. Given that the observed reactivity is unprecedented in alcohol oxidases and the suppression of the hydrolytic background reaction is difficult to achieve with more traditional transesterification (bio)catalysts, this is an unexpected and impactful finding.

The manuscript is adequately structured, and written in clear and engaging language. The introduction provides a relevant literature context, although additional articles describing applications of MsAcT (the benchmark enzyme for ester formation in water) should be cited (see below). The main findings of the present study – that PcAOx-VPN (and the analogous ParAOx-VPN and AflAOx-VPN) catalyse transesterification in aqueous medium without a hydrolytic side activity towards the ester products and that oxidised FAD and an active-site histidine residue are crucial for this catalytic activity – are firmly supported by the presented data. However, the scientific conclusions beyond these basic points, including the mechanistic proposals, are highly speculative, as they lack sufficient experimental and/or computational support. Several aspects of the reaction that are important determinants of its wider interest and impact (e.g., enantioselectivity with secondary alcohols, broader alcohol substrate scope, scope of the acid part of the acyl donor, potential hydrolytic reactivity towards the acyl donor) have not been investigated. Moreover, the manuscript and Supporting Information are strewn with inconsistencies and factual errors, and many of the experimental procedures are described so superficially that it will be difficult for other researchers in the field to reproduce them.

Overall, this submission describes an interesting finding, but lacks the scope and scientific rigour expected of work appearing in a top-tier journal such as *Nature Communications*. Many of its shortcomings are severe and can only be remedied by substantial additional work. Therefore, my recommendation is to reject the manuscript in its present form and invite resubmission only after significant extension and revision. Suggestions for improving the manuscript are provided below:

(1) Article title: The term "irreversible" in the title is somewhat misleading, as it refers to the absence of a hydrolytic side reaction, while the transesterification as such is still a reversible process (evident from the reactions in Figure 5c, which lack a strong thermodynamic driving force) if the byproduct does not undergo an irreversible follow-up reaction (as vinyl alcohol does by tautomerising to give acetaldehyde). Also, the title would be more informative if it mentioned the enzyme class studied in this work – in particular as the discovery of transesterification reactivity in alcohol oxidases is one of the most counterintuitive and hence interesting findings of the present study. I suggest "Engineered alcohol oxidases catalyse transesterification in aqueous media without competing hydrolysis" or something along these lines.

(2) p. 3, line 40: The following references for recent work with MsAcT should be added: I. C. Perdomo et al., *J. Agric. Food Chem.* 2019, 67, 6517–6522 (<https://doi.org/10.1021/acs.jafc.9b01790>); F. Annunziata et al., *Antioxidants* 2021, 10, 1142 (<https://doi.org/10.3390/antiox10071142>); B. Zdun et al., *Catalysts* 2022, 12, 1610 (<https://doi.org/10.3390/catal12121610>); A. Rudzka et al., *Angew. Chem. Int. Ed.* 2025, 64, e202420133 (<https://doi.org/10.1002/anie.202420133>); perhaps also an additional paper featuring one of the authors: N. de Leeuw et al., *Adv. Synth. Catal.* 2018, 360, 242–249

(<https://doi.org/10.1002/adsc.201701282>). For general context, a work using carboxylic acid reductases (CARs) could also be referenced, although the focus there is on the esterification of carboxylic acids with simple alcohols: P. Pongpamorn et al., *Angew. Chem. Int. Ed.* 2021, 60, 5749–5753 (<https://doi.org/10.1002/anie.202013962>).

(3) p. 5, line 72: "...its efficacy in catalysing benzyl alcohol oxidation increased over tenfold compared to the wild-type..." How were these specific activities determined? The SI provides an experimental procedure for a plate-based colorimetric assay, but no mention is made of an activity assay in solution.

(4) p. 7, line 105: "...we substituted ethyl acetate with vinyl acetate (Figure S11), which accelerated the initial reaction rate approximately twofold." Switching from ethyl acetate to vinyl acetate also increased the maximum conversion approx. twofold, hinting at a thermodynamic contribution to the increased reaction rate (irreversible reaction due to tautomerisation of byproduct). This should be discussed here.

(5) p. 7, line 108 and many other instances: Please use standard IUPAC nomenclature for referring to the studied compounds. Hence, replace "2-phenyl ethanol" with "2-phenylethanol", "2-phenylethanol acetate" (e.g., on p. 11, line 169) with "2-phenylethyl acetate", and nonsensical names such as "2-benzyl alcohol" and "(R)-1-benzyl alcohol" (SI, Figure S9) with the accepted, systematic names of these molecules.

(6) p. 8, Figure 2: The incomplete conversion attained with an equimolar amount of vinyl acetate suggests that the acyl donor – other than the ester product – might undergo hydrolysis under the reaction conditions. The fact that the two reactions with the highest alcohol concentrations (250 and 500 mM) both stall at the same product concentration (ca. 90 mM) also hints at reduced availability of acyl donor due to competing hydrolysis. The authors should investigate whether hydrolysis of vinyl acetate takes place in their reactions (e.g., by following the formation of acetaldehyde using a colorimetric assay) and if yes, whether this side reaction is enzyme-catalysed or spontaneous.

Also, in this and all other figures that visualise results of duplicate experiments, I would consider it better to show both data points for each condition individually than to show a mean plus standard deviation (error bar).

(7) p. 8, Figure 2 caption; also SI, Figures S13 and S14: Over what time period were the initial rates reported in these figures (and in the SI text) determined? Was a linear fit over several time points applied or was a single time point used for the calculation?

(8) pp. 8–9, lines 125–131, also Scheme 2: The ordered sequential mechanism shown in Cleland notation in Scheme 2 and discussed in the accompanying paragraph of text seems to be based solely on the data shown in Figure 2 (and the analogous Figure S14). With only 4–5 different concentrations of either substrate studied in the forward reaction and initial reaction rates deduced from a few end-point conversion data points, these data are neither complete enough nor accurate enough to reliably deduce a detailed order of substrate binding and product release events. More elaborate kinetic analysis, including a systematic analysis of product inhibition patterns for the forward and reverse reaction (use an acyl donor whose byproduct does not undergo tautomerisation for these experiments, e.g., ethyl acetate), is required to deduce the order of substrate binding and product release.

(9) p. 12, line 178: "...there was a clear preference towards sterically less hindered primary alcohols..." Preference compared to what? Write, for instance, "clear preference towards sterically less hindered primary alcohols compared to secondary alcohols"; line 179: "...decreased with the chain length of the alcohol..." should be changed to "decreased with increasing chain length of the alcohol"; line 182: The text mentions methoxy substituents, but Figure 5 and the chromatograms in the SI (Figure S50) show methyl substituents. Double-check and revise.

(10) p. 13, Figure 5a: A kinetic resolution reaction is shown for the secondary alcohols, in which the transesterification reaction converts the (S)-enantiomer of the alcohol to the ester and leaves the (R)-enantiomer untouched. However, no mention of enantioselectivity is made anywhere in the manuscript text or the SI, and no methods for the chromatographic separation of substrate and/or product enantiomers are reported in the analytical procedures (in fact, Figure S49 even fails to show achiral-phase separations of these compounds; see comment 27). It hence seems that the enantioselectivity in these reactions was not investigated at all. The authors should elucidate the enantioselectivity of P_cAOx-VPN (and, ideally, also of P_aAOx-VPN and A_fAOx-VPN) in transesterification reactions of sec-alcohols, as this parameter is an important determinant of the application potential of these enzymes. In general, a broader range of substrates should be investigated, including esters with acid parts other than acetyl.

(11) p. 14, Figure 6: The panels a), b) and c) in the Figure should be explained in the Figure caption.

(12) p. 15, lines 213–215, also Scheme 3: While the authors present conclusive experimental evidence supporting the direct involvement of oxidised flavin and His561 in the transesterification reaction, this is still a weak basis for proposing a detailed catalytic mechanism. What's more, the arrow-pushing mechanism drawn in Scheme 3 suggests a direct nucleophilic displacement of the alcoholate leaving group from the ester. As both enzymatic and non-enzymatic transesterification reactions are known to proceed via tetrahedral intermediates, this mechanistic proposal is very unconventional. In a way, the authors seem to be aware of this, as they have used a tetrahedral intermediate as ligand for the docking simulations described on p. S41 of the Supporting Information (Figure S20). However, these docking results are not mentioned in the manuscript text in any way. If a tetrahedral intermediate is involved, its negative charge on oxygen needs to be stabilised in some way within the enzyme's active site (cf. the "oxyanion hole" of lipases). The authors should look out for residues able to stabilise a tetrahedral intermediate in the crystal structure of the enzyme and in the docking poses. They should also investigate whether the reaction is inhibited by hydrolase inhibitors of the phosphonate type, which mimic the tetrahedral

intermediate. On the basis of the currently presented data alone, the mechanistic proposal is highly speculative.

(13) p. 15, line 223: "...and likely have a similar active site architecture." Why make this speculative statement here? The multiple-sequence alignment shown in Figure S8 (which is barely discussed in the manuscript text, and misnumbered as Figure S17) contains all the information needed to compare the active-site residues of the three enzymes. For instance, it shows residues W560, H561, and N604 to be fully conserved. Also, with such a high sequence similarity between the three enzymes, it would be easy to generate high-quality homology models of ParAOx and AflAOx to compare their active sites to that of PcAOx on a 3D-structural basis. Finally, the authors speak of "sequence similarity" in the text, but I assume the percentages given in parentheses refer to sequence identity. This should be made clear, e.g.: (ParAOx: 87% identity, AflAOx: 53% identity).

(14) p. 16, line 228: "Figures S17, S18" I believe the authors want to refer to Figures S4 and S5 here. In general, misnumbering of Figures in cross-references is a common problem with this manuscript. Double-check all cross-references and revise.

(15) SI, p. S4: "...by adding 1 μL Dpn I enzyme for every 1 μg PCR product." Please specify the concentration and/or the specific activity of the DpnI enzyme solution used. "...containing 50 $\mu\text{g}\times\text{mL}^{-1}$ at 37 $^{\circ}\text{C}$..." Containing 50 $\mu\text{g}/\text{mL}$ of what? I assume kanamycin.

(16) SI, p. S5: "...oxidative coupling of phenol and 4-amino antipyrine..." No mention of phenol in the screening medium is made, instead 3,5-dichloro-2-hydroxybenzenesulfonic acid is mentioned as the coupling partner. Double-check and revise the text and Figure S1. "After 30 h cultivation at 30 $^{\circ}\text{C}$, single red colonies were selected for sequencing." Please provide more details on the resulting variant library: How many positives were sequenced? How many of them were wild types? What variants were found in the pool of sequenced positives?

(17) SI, p. S6: "Crude cell extracts were prepared..." Please provide further experimental details of the cell disruption and purification procedure: How much buffer was used for what quantity of cell pellet? What types of Ni column and desalting column were used (dimensions, manufacturer; carrier material for the Ni column: I assume Ni-NTA)? Was elution performed with a linear gradient over the specified imidazole concentration range or was a step-elution protocol used? How many column volumes of eluent were used for elution? Was the chromatography performed manually (e.g., with a syringe or a peristaltic pump) or using an FPLC instrument?

(18) SI, pp. S6–S8: Please use precise and consistent terminology to refer to the bands in the SDS-PAGE images. "the total bacteria" and "whole cell protein" are not precise, nor are they consistent with the terminology used in Figure S2, where "cell extract of *E. coli* overexpressing PcAOx" is used. Also, "250 mM imidazole eluting pure enzyme" in the caption of Figure S5 is a poor description, as this protein fraction is clearly not pure, with many additional bands visible.

(19) SI, p. S8: "...was further purified using gel filtration chromatography..." Please provide experimental details: What type and dimension of column were used? What was the mobile phase? What was the volume and concentration of applied (crude) protein solution? What was the volume of the collected fractions?

(20) SI, p. S12: "The concentrations of reagents and enzymes as well as reaction conditions are specified in the figure captions." Please provide further experimental details: I assume that stock solutions were used to prepare the reaction mixtures. What were their concentrations and in what order were they mixed to prepare the reactions? "All quantifications are based on calibration curves using authentic standards." Was the quantification performed relative to an internal standard or based directly on the analyte integrals?

(21) SI, p. S13, Figure S8: To make this figure more informative, the residues that were mutated to generate transesterification activity and other active-site residues should be highlighted.

(22) SI, p. S34, Figure S12: How was the residual activity determined (type of assay, reaction time)? "10% (w/v) PMFS" The abbreviation is PMSF, not PMFS. 10% w/v = 100 g/L = 0.574 mol/L. This concentration is too high to be plausible.

(23) SI, p. S35, caption of Figure S13: The dot markers for benzyl acetate and the dashed line for the sum of ester and aldehyde should be shown in red colour, as they are in the Figure itself.

(24) SI, p. S38, Figure S17: The figure misses a legend that explains what the three differently coloured bars refer to.

(25) SI, pp. S39–S40, captions of Figures S18–S19: How were the temperature tolerance and pH tolerance of the enzymes determined? By pre-incubation of the enzyme solution at the indicated temperature (and pH) for the indicated time? How was activity determined (type of assay, reaction time)?

(26) SI, p. S41: Please provide information on how the enzyme structures were prepared for docking (e.g., adjustment of protonation state, calculation of charges, setup of simulation cell).

(27) SI, pp. S43–S76, gas chromatograms: Chromatograms of reactions using high substrate concentrations show heavy fronting, a sign of exceeding the capacity of the GC column. The quantifications based on these chromatograms are likely imprecise. The samples should have been diluted before analysis. Also, the cross-references to the Figures in the

manuscript are incorrect: For instance, "Chromatograms related to Figure 2" seem to refer to a Figure that is no longer part of the manuscript, "Chromatograms related to Figure 3" actually refers to Figure 2, etc. Double-check and revise. The chromatograms shown on pp. S55 and S56 cannot possibly correspond to the given reactant concentrations ([vinyl acetate] = 50 mM, [2-phenylethanol] = 250 mM), as the final two time points show complete consumption of the alcohol, which is not possible with sub-stoichiometric amounts of acyl donor. "Chromatograms related to Figure 4" again do not refer to the correct figure. Also, why is there heavy tailing now for a chromatographic separation that gave good peak shape in earlier chromatograms? In Figure S49 (pp. S66–S67), only primary alcohols (and unbranched esters) are shown, in contrast to what the figure caption claims.

(28) Typos: p. 5, line 68: alcohol → alcohols; p. 6, Figure 1 caption: yellow → white, W590 → W560; p. 7, line 114: effect the → effect of the; p. 7, Figure 2: 1-phenylethanol acetate → 2-phenylethyl acetate, 1-phenyl ethanol → 2-phenylethanol; p. 9, line 130: doner → donor; p. 11, line 160: upon completion of the activate acyl donor → upon complete consumption of the activated acyl donor; p. 12, line 183: these → this; p. 12, line 185: in all cases → in both cases; p. 14, line 204: catalyts → catalyst; p. 15, line 217: His601 → His561; p. 15, line 218: it → its; SI, p S12: Biotansformations → Biotransformations; SI, p S12: cures → curves; SI, p. S31: compared and → and; SI, p. S36: 2-phenyl ethanol → benzyl alcohol; SI, p. S67: china → chain

Version 1:

Reviewer comments:

Reviewer #1

(Remarks to the Author)
Dear Authors,

In general I am satisfied with the revision; however, I have a few additional minor remarks as stated below:

- (1) Please erase the comment on page S12 in Supporting Information.
- (2) The catalytic channel/tunnel depicted in Figure S1 could be better visualized (e.g., a larger picture is needed in general, including the measured diameter of the channel at the entrance and in the bottlenecks, as well as a more in-depth discussion on the pronounced hydrophobic amino acid characteristics).
- (3) Please add the discussion to Figure S2, as it is unclear to me. What does the middle line mean, except as a timescale for the reaction? Is it the interface of the external reaction medium and the enzyme active site? One can guess what all these mean, but it should be explicitly specified/clarified.
- (4) The authors investigated the enantioselectivity of the PcAOx-VPN-catalyzed esterification of 1-phenylethanol and lactic acid using both enantiomers separately and as racemates; however, they did not present the results for E-values (or I missed something?). Please comment on this.
- (5) The authors show PcAOx crystal structures for wild-type PcAOx and engineered variant PcAOx-VPN in Figure S3, and speak about some angles "a and b" between these two enzymes and the W560 residue. I was slightly puzzled when I saw no angles and their values. Please specify (visualized) with dashed lines which of these angles you are speaking about, and calculate their values if it is so important. Otherwise, it is too speculative, and it adds absolutely nothing to this research.
- (6) The experimental procedure describing the preparative-scale reaction presented in Section 2.12 of the Supporting Information is concise and should be revised to include more details to ensure full reproducibility for other researchers. For example, what does it mean that the product was distilled? Please specify the conditions in terms of the pressure and temperature. I also see no data concerning the isolated yield that is given in the main manuscript, but why is it not included in the SI document?
- (7) In the previous round of revision, I asked authors the following: "(...) In the 'Supporting Information' file, I recommend compiling all relevant details regarding the recombinantly produced enzymes into a single table. This table should include information on the vector and expression host used, plasmid number, plasmid vector, and the inducer used for protein expression. (...)". To my great disappointment, I do not see the table with the collected information provided in SI. Please complete this.
- (8) Figure 4 in the main manuscript contains blurred chemical structures. I recommend avoiding the use of coloring compounds on the scheme in this style, especially when using red and blue highlights.

Respectfully yours,
Anonymous Reviewer

Reviewer #2

(Remarks to the Author)

The authors of this work presented the discovery of a mutant of alcohol oxidase from *Phanerochaete chrysosporium* (PcAOx-VPN) whose more hydrophobic cavity is instrumental to avoid hydrolytic side activity in transesterification reactions. As stated in the first round of review, I think that this is a well-done and relevant piece of work. All issues were addressed by authors.

Reviewer #3

(Remarks to the Author)

With their revised manuscript, Ma, Hollmann, Wang, and co-workers show an honest and substantial effort to improve their paper based on the criticism raised by myself and the other reviewers of the original submission. The authors address every reviewer comment openly and adequately, they report a substantial body of additional experimental work, which clarifies many of the questions that the original manuscript left unanswered and thereby significantly improves the overall quality of the paper, and they have incorporated numerous small changes to the text and figures to raise the overall level of clarity and data presentation. Most importantly, the authors now demonstrate a significantly broader substrate scope than in their original submission, including non-alcohol nucleophiles and a more diverse set of acyl donors, and they provide convincing experimental evidence for (S)-stereoselectivity with chiral alcohol substrates. The mechanistic proposal has also been amended, and it is now communicated as a working hypothesis for more detailed enzymological studies to be carried out in the future.

With these significant improvements, I can now recommend this work for publication in Nature Communications. A few minor points still require correction, but I am optimistic that these can be handled without further peer review. Suggestions for final improvements and corrections are provided below:

(1) Abstract: Some of the minor text changes made during revision seem to have altered or obscured the originally intended meaning, and a few passages have been unclear even in the original: "but they are less elegant" – less elegant compared to what alternatives? // "In this study, we present a mutant of alcohol oxidase from *Phanerochaete chrysosporium* (PcAOx-VPN) that exhibits no hydrolytic activity." – At this point in the abstract, the (unexpected) transesterification activity of PcAOx-VPN has not yet been introduced, so the statement is lacking context. // "...enabling transesterification reactions in water, previously considered impossible, ..." – Transesterification in water has been demonstrated to be possible, even if it has been plagued by competing hydrolysis. Please revise these passages for improved clarity.

(2) Scheme 1 caption: "Comparison of the time courses of classical transesterification time courses..." I think the second instance of "time courses" should be deleted.

(3) Non-standard nomenclature: p. 6, line 98: 1,2-propanol → propane-1,2-diol, line 100: propane diol → propanediol; p. 14, line 225: 1-phenyl ethanol → 1-phenylethanol

(4) p. 9, line 149: "As enzyme inactivation..." This sentence seems to lack a verb.

(5) p. 13, line 199: "This is consistent with Fraaije's observation that..." Because of text insertions, the cross-reference using "this" has become ambiguous. Better re-phrase, e.g. "The absence of hydrolysis activity is consistent..."

(6) p. 13, line 207: "Also benzyl acetate and 2-phenylethyl acetate were accepted as acyl donors (Figure 5, [b] and [c])." Since the authors discuss restrictions in active-site access for bulky substrates just a few lines above, it would help to clarify that benzyl acetate and phenethyl acetate are accepted as acyl donors only when combined with small, aliphatic alcohols as substrates.

(7) p. 13, line 211: "Amongst the aliphatic alcohols tested, primary alcohols (4-8) were preferred..." In Figure 5, substance numbers 4–8 refer to the ester products, not the alcohol substrates. Although it is clear what the authors want to say, I would consider it advantageous to develop a clear and systematic abbreviation scheme that allows alcohols, acyl donors, and products to be unambiguously referred to, e.g., by using numbers for alcohols, letters for acyl donors, and a combination of number and letter for the resulting ester products.

(8) p. 18, lines 277–287, also Scheme 2 and Figure S8: What the authors call a "transition state" is really an intermediate and should be referred to accordingly. [Note that a transition state represents a state of maximum energy in a reaction trajectory, where bonds are partly formed and broken. (cf. the IUPAC definition of 'transition state': <https://goldbook.iupac.org/terms/view/T06468>) In contrast, the tetrahedral intermediate shown in Scheme 2 and referred to in the text is a chemical entity that represents a local energy minimum and comprises only fully formed bonds, hence why it is called an 'intermediate' (<https://goldbook.iupac.org/terms/view/I03096>).] As intermediates are not usually drawn in brackets, I suggest to remove the brackets from Scheme 2b. Also, the arrow pushing in Scheme 2b is only partially correct, as not all bonding changes occurring between the shown states are accounted for by curly arrows (for instance, the "flipping" of the double bond in the histidine's imidazole ring is not accounted for). Please formulate a complete arrow-pushing mechanism. Besides, in Figure S8, all four structures miss an oxygen atom (the one of the alcohol part of the acyl donor).

(9) SI, Section 1.6: Please provide instrumentation and conditions for the chiral analyses that have been carried out during revision. Please also provide exemplary chromatograms of enantiomeric separations (similar to those shown in Section 2.10) in a format that clearly demonstrates separation of a racemic standard under the same conditions used for separation of the biotransformation samples. If any of the samples were derivatised before analysis (I consider this likely for the carboxylic acid derivatives 28 and 29), please provide the derivatisation procedure that was used.

(10) SI, Section 1.7: This section now contains a description of a molecular docking workflow that is even less informative than the corresponding paragraph in the original submission. On the other hand, a paragraph on docking that provides adequate details of the workflow is provided in Section 2.2. Please combine the two paragraphs into one and clarify which of the two workflows was used for what docking problem. Also, the SI Word file still contains author comments in Section 1.7 that the authors might want to remove before publication.

(11) Typos: p. 11, line 190: "may not sufficiently spacious" → "may not be sufficiently spacious", p. 20, line 317: "We therefore hypothesis" → "We therefore hypothesise"

Detailed answer to the reviewers' comments:

Reviewer #1

The submitted manuscript, entitled "Irreversible transesterification reactions in aqueous media" (NCOMMS-24-71991-T), authored by F. Hollmann, Y. Wang, et al., presents the discovery of a novel engineered variant of alcohol oxidase from *Phanerochaete chrysosporium* (PcAOx-VPN) exhibiting unexpected ('promiscuous') quasi-irreversible transesterification activity in water, without reversed hydrolytic action. The reported biocatalyst is the first known flavin-containing enzyme capable of catalyzing a reaction beyond its native catalytic repertoire. As such, it has the potential to serve as a superior alternative to existing variants of *Mycobacterium smegmatis* acyltransferase (MsAcT).

I find this paper highly interesting and valuable to the broader scientific community, as the reported research presents transformative results that significantly advance the field of bioorganic chemistry. Moreover, it has the potential to open new avenues for future synthetic endeavors, particularly in the transesterification of alcohols. In this context, the study exemplifies the critical need for discovering novel biocatalysts with unnatural features that enable challenging transformations—often unattainable through classical chemical methodologies due to strongly unfavorable thermodynamics. A key strength of this paper is the successful development of the efficient triple mutant M59V-Q60P-R61N (PcAOx-VPN) through saturation mutagenesis. Notably, this variant demonstrated the ability to catalyze the desired transesterification of selected model primary alcohols with bulkier aryl substituents, such as benzyl alcohol and 2-phenylethanol.

However, while the newfound activity of the M59V-Q60P-R61N variant is highly promising, there are notable limitations in the tested substrate scope. The study suggests that the method is restricted to a narrow range of substrates, predominantly specific primary alcohols. Interestingly, PcAOx-VPN also catalyzes the efficient alcoholysis of benzyl esters using various aliphatic alcohols (MeOH, EtOH, n-PrOH, and n-BuOH), which could serve as a promising alternative to hydrolytic processes used in applications such as biodiesel production.

The experimental and analytical methods were rigorously designed and executed to meet high standards, ensuring reliability and reproducibility. A thorough analysis of the "Supporting Information" file confirms that the results are well-supported by appropriately designed controls, detailed analyses, and robust statistical evaluation, with no significant flaws or oversights. Technically, the reported data are well-documented, and the paper is written in a clear, concise, and accessible manner. Furthermore, all included figures, tables, and supplementary materials are of high quality, well-organized, and effectively communicate the results. Given its exceptional quality, novelty, and significant impact, I recommend that this manuscript be considered for publication in *Nature Communications*. However, I suggest that the following issues be carefully addressed by the authors prior to publication:

Thank you very much for sharing our enthusiasm on *PcAOx-VPN* and for the valuable suggestions, which helped us to improve the quality of the manuscript.

Minor points:

(1) In general, the title of the submitted manuscript is slightly misleading, and more suitable for review-type article rather than a communication in Nature journal. In general, it suggests as it report on this topic in a broader context. In my opinion, the title should relate more to discovered variant of alcohol oxidase from *Phanerochaete chrysosporium* (*PcAOx-VPN*) and its promiscuous catalytical activity in terms of transesterification in water.

We fully agree with the reviewer and have changed the title to 'Engineered alcohol oxidases catalyse transesterification in aqueous media without competing hydrolysis'.

(2) In the 'Introduction' section authors stated that: "(...) Serendipitously, we discovered transesterification activity in variants of the alcohol oxidase from *Phanerochaete chrysosporium* (*PcAOx*)11-13 (...)". After careful reading of all three cited articles, I cannot confirm that any of the authors from the list of the submitted paper participated in those discoveries. Therefore, please rephrase this sentence as it suggests that the authors took part in those studies. As I understand correctly, Prof. Marco W. Fraaije (who is one of the authors of the papers cited in Refs. 11–13) is affiliated with the University of Groningen (NL) and not with TU Delft (NL).

We thank the reviewer for pointing out this inaccuracy, which we have, of course, corrected. The original contributions of the Fraaije group have now been emphasized more clearly.

(3) In the 'Introduction' section, there is an insufficient number of references concerning the application of *Mycobacterium smegmatis* acyltransferase (*MsAct*). I strongly recommend adding some key citations that highlight the potential of this enzyme in synthetic applications, such as: (i) P. Borowiecki et al. *Angew. Chem. Int. Ed.* 2025, 64, e202420133; (ii) U. T. Bornscheuer et al. *ACS Catal.* 2021, 11, 14906–14915; (iii) U. T. Bornscheuer et al. *Angew. Chem. Int. Ed.* 2020, 59, 11607–11612; (viii) W. Kroutil and F. Himo et al. *ACS Catal.* 2018, 8, 10698–10706; (ix) U. Hanefeld et al. *Adv. Synth. Catal.* 2017, 360, 242–249, etc.

Following the reviewer's suggestion, we have added the relevant literature references.

(4) Please refine the following sentences: "(...) The incubation of for example 2-phenylethanol acetate with *PcAOx-VPN* (...)". The chemical name of "2-phenylethanol acetate" is faulty in this case. It should be "2-phenylethyl acetate".

Corrected as pointed out by the reviewer.

(5) Out of curiosity, I would like to ask the authors whether they have tested other quasi-irreversible acyl donors (e.g., fatty acid-like vinyl esters, isopropenyl acetate, acetic anhydride, 4-nitrophenyl acetate, 4-chlorophenyl acetate, 2,2,2-trifluoroethyl acetate, etc.).

Following the reviewer's suggestion we have tested these alternative acyl donors. The results are shown in the revised SI and a brief discussion has been added to the manuscript.

(6) The authors stated that: "(...) For the acyl donor vinyl acetate, however, substrate inhibition was detected (...)" and then: "(...) We interpret these observations as a result of an ordered, sequential reaction mechanism (...)". Maybe the drop in the catalytic activity is due to the in situ formation of acetaldehyde. As this might be true, once again, I suggest testing other acyl donors which does not generate acetaldehyde byproduct.

We thank the reviewer for this useful suggestion and have consequently performed additional experiments now included in the revised manuscript and SI. Indeed the autohydrolysis rate of vinyl acetate under reaction conditions was significantly higher than we originally expected (ca. 0.5 mM h⁻¹). With this in mind, of course also acetaldehyde-related inactivation cannot be excluded as cause of the apparent substrate inhibition. Therefore, we also performed experiments with isopropenyl acetate as acyl donor (now included in the revised manuscript and SI). We found that, while exhibiting a somewhat lower autohydrolysis rate and producing a far less noxious byproduct (acetone instead of acetaldehyde), similar inhibition patterns were observed. We believe that these results support our suggested competitive substrate inhibition mechanism.

(7) The authors present the results of biotransformations using concentrations of the product obtained. I think it would be better to show a conversion (% conv.).

We thank the reviewer for this helpful suggestion. We have implemented it in the substrate scope figure, where GC-yields are now shown as recommended. For the other figures, we believe that the measured product concentrations provide more informative and relevant data in the context of our study. We hope this rationale is acceptable.

Major points:

- (1) This study is significantly limited by its narrow substrate scope, as it primarily focuses on ortho-, meta-, and para-substituted benzyl alcohols. In particular, the absence of racemic secondary alcohols is a major drawback, as their inclusion could provide valuable insights into the catalytic capabilities of the engineered variant, especially regarding enantioselectivity. To address this limitation, I strongly recommend revising the manuscript by applying the PcAOx-VPN variant to the kinetic resolution of representative racemic secondary alcohols. These should include benzylic alcohols, aliphatic alcohols, "small-bulky" aliphatic-aryl alcohols, hydroxy esters, heteroaromatic (halo)methyl alcohols, and "bulky-bulky" aryl-aryl alcohols. Furthermore, all reactions and the resulting optically active products should be fully characterized in terms of their stereochemical outcomes. This should include an assessment of the enzyme's stereopreference, the enantiomeric excess (% ee) of the isolated kinetic resolution (KR) products, and the enantioselectivity (E-value).
- (2) I also strongly encourage the authors to test their enzyme on substrates containing more than a single hydroxyl group. Using exemplary diols, triols (such as simple glycerol), and other benchmark

multi-hydroxyl compounds could help demonstrate the regioselectivity of the investigated biocatalyst. Additionally, the inclusion of at least one amino alcohol as a substrate would be valuable in assessing the chemoselectivity of the discovered biocatalyst.

We thank the reviewer for this valuable suggestion and have now investigated the enantioselectivity of *PcAOx-VPN* towards various chiral alcohols. These results have been added to the revised supplementary information and are mentioned in the updated manuscript. We also appreciate the reviewer's extensive suggestions regarding possible racemic starting materials. However, we hope the reviewer will understand that the primary focus of this study is the newly discovered transesterification activity (not previously observed with flavin-dependent enzymes) and the apparent absence of hydrolytic activity in *PcAOx-VPN*. Further substrate exploration and a detailed kinetic characterisation are planned for future studies.

(3) The authors stated that the unusual ability of *PcAOx-VPN* to catalyze transesterification in water without hydrolysis is attributed to the hydrophobic architecture of its active site. To support this hypothesis regarding the catalytic mechanism of action, I strongly recommend visualizing the tunnels and channels in the molecular structure of *PcAOx-VPN* that lead to the active site entrance. This can be achieved using software such as CAVER Analyst 2.0, developed by researchers at the Loschmidt Laboratories and the National Centre for Biomolecular Research (NCBR) at Masaryk University. Such an analysis could provide deeper insights into the structural factors and molecular basis that determine the rate-limiting step, particularly concerning diffusion effects. Additionally, it would enhance the rationalization of the observed substrate scope.

We thank the referee for this helpful suggestion. The tunnel analysis has now been included in the SI and is discussed in the revised manuscript.

(4) Since Nature Communications values research that bridges disciplines and has applications across multiple fields, I recommend that the authors enhance their paper to align with these standards, ensuring that their work resonates with a broader scientific audience beyond specialists in biocatalysis. To achieve this, I suggest highlighting potential applications of the transesterification methodology in a broader context. For instance, demonstrating its extension toward the synthesis of exemplary small-molecule pharmaceuticals, bioactive compounds, or specialty chemicals would significantly strengthen its relevance. Additionally, this would help justify the selection of model alcohols. Without such examples, the impact of this methodology may be perceived as limited to a narrowly defined set of substrates, potentially restricting its broader applicability and diminishing its potential influence on future discoveries in modern bioorganic chemistry.

Thank you very much for this valuable and constructive suggestion. In response, we have expanded the discussion to better highlight the potential applications of our transesterification methodology in a broader context. To further illustrate the synthetic utility of the enzyme, we have also included a preparative-scale transformation, demonstrating its relevance beyond analytical-scale experiments. At the same time, we would like to emphasise the novelty of this work. To the best of our knowledge, this is the first report of a flavoenzyme catalysing a transesterification reaction. The ability to

perform transesterification in the absence of competing hydrolysis (rather than merely diminished hydrolytic activity as in the case of *MsAcT*) is a particularly striking feature. We believe this finding will be of interest not only to enzymologists, but also to protein engineers and synthetic chemists. Taken together, we are convinced that this work will resonate with a broad scientific readership and provide a foundation for future applications in biocatalysis and beyond.

(5) In the "Supporting Information" file, I recommend compiling all relevant details regarding the recombinantly produced enzymes into a single table. This table should include information on the vector and expression host used, plasmid number, plasmid vector, and the inducer used for protein expression.

Following the reviewer's suggestion we have clarified the expression hosts, expression vectors and expression conditions in the SI.

(6) According to the Notice for Authors set by prestigious journals: "(...) For synthetic methods, a selection of substrates illustrative of the scope of the reaction should be made, and the isolated yields of the corresponding purified products should be reported. Product yields determined by HPLC or GC are considered indicative of a preliminary catalytic study (...)." Therefore, to better illustrate the applicability of the method, I strongly recommend including at least one reaction performed on a gram scale.

Following the reviewer's suggestion, we have performed a gram-scale reaction, now included in the revised manuscript and SI.

Reviewer #2

This manuscript describes a protein engineering work on alcohol oxidase from *Phanerochaete chrysosporium* (PcAOx) which led to the identification of a triple mutant that is able to perform a transesterification reaction avoiding undesired hydrolysis occurring with other enzymes. PcAOx can be expressed in *E. coli* and already proved to be prone to rational modification of substrate specificity from alcohols to glycerol, representing an interesting biocatalytic tool. This work is relevant and could have interesting implications, nevertheless I have some issues:

- crystal structure of PcAOx-VPN: a figure showing the electron density of the active site (for example, the map of the FAD cofactor and that of the mutated residues) must be included. Moreover, Table S2 is reporting an R-merge of 27%, 269% in the highest resolution shell: this is quite high whereas the authors declare a CC1/2 of 0.998 (0.720 in the highest resolution shell). How do you explain this? Why the R-merge/R-sym was not reported during PDB deposition as gathered from the table at page 8 of the PDB validation report?

We sincerely appreciate your pointing out the issues in our data processing, your input has been extremely helpful. We mistakenly used the results of automated data processing in the analysis of this structure. The resolution cutoff was set too high, resulting in the inclusion of lower-quality data from the highest resolution shell, which ultimately led to the appearance of outliers. To rectify this error and improve the quality of the crystallographic data, we manually reprocessed the original dataset, optimised the final structural model, and adjusted the resolution from 2.5 Å to 2.9 Å. The revised model has been re-uploaded to the PDB database (PDB ID: 9V6K), and we have updated the data report accordingly, including the requested values in the validation report.

The optimised model remains largely unchanged overall, with only minor adjustments in specific regions. Corresponding revisions have been made in the relevant sections of the manuscript. In addition, we fully agree with your suggestion and have implemented it by including electron density maps for key sites in the appropriate locations, providing clearer support for our interpretation.

- lines 76-77: Figures S12 and S13 are not related to crystal structures.

Corrected.

- I would suggest to include the term alcohol oxidase in the title because the serendipitous discovery of an alcohol oxidase performing a transesterification reaction is an important aspect of this work.

Changed as suggested by the reviewer.

- Fig. S3: the spectrum shows that the 359 nm is (besides the gap in Abs measurement probably due to reading problems) much higher than the peak at 457 nm, which is unusual in flavoproteins. Although the spectrum is not extended to the UV range, the Abs baseline seems to increase from 700 nm backwards, which is typical of turbidity. Indeed, the protein sample in the picture looks a bit turbid. Can authors comment on this aspect?

We apologise for having provided low-quality spectra due to precipitation, as rightly pointed out by the reviewer. We have re-recorded the UV/Vis spectrum of our *PcAOx*-VPN and included it in the revised Supporting Information.

- Supplementary page S4: authors should specify what they mean with the primer portions highlighted in yellow. This appears also in other sections of the Supplementary text. One can guess what they mean, but it should be explicitly specified.

Corrected as suggested by the reviewer.

Reviewer #3

The manuscript submitted by Hollmann, Wang, and co-workers reports the (serendipitous) discovery that a triple variant of alcohol oxidase from *Phanerochaete chrysosporium* (PcAOx-VPN) catalyses transesterification reactions of several simple primary and secondary alcohols with vinyl acetate (and a few additional acetyl esters) in aqueous medium, notably without a detectable hydrolysis side activity. The authors explore the scope of this transformation, show that it can be generalized to other alcohol oxidases, and make some proposals regarding its catalytic mechanism. Given that the observed reactivity is unprecedented in alcohol oxidases and the suppression of the hydrolytic background reaction is difficult to achieve with more traditional transesterification (bio)catalysts, this is an unexpected and impactful finding.

The manuscript is adequately structured, and written in clear and engaging language. The introduction provides a relevant literature context, although additional articles describing applications of MsAcT (the benchmark enzyme for ester formation in water) should be cited (see below). The main findings of the present study – that PcAOx-VPN (and the analogous ParAOx-VPN and AflAOx-VPN) catalyse transesterification in aqueous medium without a hydrolytic side activity towards the ester products and that oxidised FAD and an active-site histidine residue are crucial for this catalytic activity – are firmly supported by the presented data. However, the scientific conclusions beyond these basic points, including the mechanistic proposals, are highly speculative, as they lack sufficient experimental and/or computational support.

Several aspects of the reaction that are important determinants of its wider interest and impact (e.g., enantioselectivity with secondary alcohols, broader alcohol substrate scope, scope of the acid part of the acyl donor, potential hydrolytic reactivity towards the acyl donor) have not been investigated. Moreover, the manuscript and Supporting Information are strewn with inconsistencies and factual errors, and many of the experimental procedures are described so superficially that it will be difficult for other researchers in the field to reproduce them. Overall, this submission describes an interesting finding, but lacks the scope and scientific rigour expected of work appearing in a top-tier journal such as *Nature Communications*. Many of its shortcomings are severe and can only be remedied by substantial additional work. Therefore, my recommendation is to reject the manuscript in its present form and invite resubmission only after significant extension and revision. Suggestions for improving the manuscript are provided below:

Thank you very much for sharing our enthusiasm on PcAOx-VPN! We hope that the additional experimental evidence and refined manuscript will convince the reviewer.

(1) Article title: The term "irreversible" in the title is somewhat misleading, as it refers to the absence of a hydrolytic side reaction, while the transesterification as such is still a reversible process (evident from the reactions in Figure 5c, which lack a strong thermodynamic driving force) if the byproduct does not undergo an irreversible follow-up reaction (as vinyl alcohol does by tautomerising to give acetaldehyde). Also, the title would be more informative if it mentioned the enzyme class studied in this work – in particular as the discovery of transesterification reactivity in alcohol oxidases is one of the most counterintuitive and hence interesting findings of the present study. I suggest

"Engineered alcohol oxidases catalyse transesterification in aqueous media without competing hydrolysis" or something along these lines.

We thank the reviewer for pointing this out. Indeed the transesterification reaction is not irreversible as pointed out by the reviewer. Following the reviewer's suggestion we have changed the title to 'Engineered alcohol oxidases catalyse transesterification in aqueous media without competing hydrolysis'

(2) p. 3, line 40: The following references for recent work with MsAcT should be added: I. C. Perdomo et al., J. Agric. Food Chem. 2019, 67, 6517–6522 (<https://doi.org/10.1021/acs.jafc.9b01790>); F. Annunziata et al., Antioxidants 2021, 10, 1142 (<https://doi.org/10.3390/antiox10071142>); B. Zdun et al., Catalysts 2022, 12, 1610 (<https://doi.org/10.3390/catal12121610>); A. Rudzka et al., Angew. Chem. Int. Ed. 2025, 64, e202420133 (<https://doi.org/10.1002/anie.202420133>); perhaps also an additional paper featuring one of the authors: N. de Leeuw et al., Adv. Synth. Catal. 2018, 360, 242–249 (<https://doi.org/10.1002/adsc.201701282>). For general context, a work using carboxylic acid reductases (CARs) could also be referenced, although the focus there is on the esterification of carboxylic acids with simple alcohols: P. Pongpamorn et al., Angew. Chem. Int. Ed. 2021, 60, 5749–5753 (<https://doi.org/10.1002/anie.202013962>).

We thank the reviewer for this extensive and relevant list of references. We have included all of the suggested citations, with the exception of the example involving ATP-driven esterification. We felt that this particular reference falls somewhat outside the scope of the present manuscript, and that providing similarly comprehensive coverage of the literature on ATP-driven esterification, as we have now done for MsAcT, would overly expand the reference list.

(3) p. 5, line 72: "...its efficacy in catalysing benzyl alcohol oxidation increased over tenfold compared to the wild-type..." How were these specific activities determined? The SI provides an experimental procedure for a plate-based colorimetric assay, but no mention is made of an activity assay in solution.

We apologise for the omission of some experimental procedures such as the UV/Vis-based activity assay. This information (and further) have now been included in the revised Supporting Information.

(4) p. 7, line 105: "...we substituted ethyl acetate with vinyl acetate (Figure S11), which accelerated the initial reaction rate approximately twofold." Switching from ethyl acetate to vinyl acetate also increased the maximum conversion approx. two-fold, hinting at a thermodynamic contribution to the increased reaction rate (irreversible reaction due to tautomerisation of byproduct). This should be discussed here.

We thank the reviewer for this insightful observation. We agree that the increase in maximum conversion upon switching from ethyl acetate to vinyl acetate suggests a thermodynamic contribution to the observed rate enhancement. As now discussed in the revised manuscript, the use of vinyl acetate likely provides a greater thermodynamic driving force due to the irreversible product

removal. We have added a corresponding remark in the text to clarify this point.

(5) p. 7, line 108 and many other instances: Please use standard IUPAC nomenclature for referring to the studied compounds. Hence, replace "2-phenyl ethanol" with "2-phenylethanol", "2-phenylethanol acetate" (e.g., on p. 11, line 169) with "2-phenylethyl acetate", and nonsensical names such as "2-benzyl alcohol" and "(R)-1-benzyl alcohol" (SI, Figure S9) with the accepted, systematic names of these molecules.

Corrected as indicated by the reviewer.

(6) p. 8, Figure 2: The incomplete conversion attained with an equimolar amount of vinyl acetate suggests that the acyl donor – other than the ester product – might undergo hydrolysis under the reaction conditions. The fact that the two reactions with the highest alcohol concentrations (250 and 500 mM) both stall at the same product concentration (ca. 90 mM) also hints at reduced availability of acyl donor due to competing hydrolysis. The authors should investigate whether hydrolysis of vinyl acetate takes place in their reactions (e.g., by following the formation of acetaldehyde using a colorimetric assay) and if yes, whether this side reaction is enzyme-catalysed or spontaneous. Also, in this and all other figures that visualise results of duplicate experiments, I would consider it better to show both data points for each condition individually than to show a mean plus standard deviation (error bar).

Following the reviewer's suggestion, we investigated the autohydrolysis of vinyl acetate by NMR (now included in the revised Supporting Information). Using 250 mM vinyl acetate, we estimated a hydrolysis rate of approximately 0.5 mM h^{-1} . No significant difference was observed between experiments conducted in the presence or absence of *PcAOx*. As autohydrolysis accounts for only minor depletion of vinyl acetate ($< 10 \text{ mM}$, i.e. $< 4\%$), we consider it an unlikely explanation for the reviewer's observation. Instead, we suspect that the limited intrinsic stability of *PcAOx*-VPN in the presence of high substrate concentrations as a likely cause.

Regarding the reviewer's suggestion to display all individual data points rather than averages with standard deviations, we found that this approach made the graph overly cluttered. We have therefore chosen not to adopt this suggestion, in order to maintain clarity and readability.

(7) p. 8, Figure 2 caption; also SI, Figures S13 and S14: Over what time period were the initial rates reported in these figures (and in the SI text) determined? Was a linear fit over several time points applied or was a single time point used for the calculation?

Wherever possible, initial rates were determined by linear regression of the linear portion of the time course. In cases where no clear linearity was observed, the estimation was based on the first time point. This has been clarified in the revised manuscript text.

(8) pp. 8–9, lines 125–131, also Scheme 2: The ordered sequential mechanism shown in Cleland notation in Scheme 2 and discussed in the accompanying paragraph of text seems to be based solely on the data shown in Figure 2 (and the analogous Figure S14). With only 4–5 different concentrations

of either substrate studied in the forward reaction and initial reaction rates deduced from a few end-point conversion data points, these data are neither complete enough nor accurate enough to reliably deduce a detailed order of substrate binding and product release events. More elaborate kinetic analysis, including a systematic analysis of product inhibition patterns for the forward and reverse reaction (use an acyl donor whose byproduct does not undergo tautomerisation for these experiments, e.g., ethyl acetate), is required to deduce the order of substrate binding and product release.

We agree with the reviewer that the proposed mechanistic model is not well supported by experimental evidence. We have therefore removed the corresponding scheme and have revised the text to emphasise more clearly the hypothetical nature of the proposed ordered sequential mechanism.

(9) p. 12, line 178: "...there was a clear preference towards sterically less hindered primary alcohols..." Preference compared to what? Write, for instance, "clear preference towards sterically less hindered primary alcohols compared to secondary alcohols"; line 179: "...decreased with the chain length of the alcohol..." should be changed to "decreased with increasing chain length of the alcohol"; line 182: The text mentions methoxy substituents, but Figure 5 and the chromatograms in the SI (Figure S50) show methyl substituents. Double-check and revise.

We thank the referee for the helpful suggestion and for pointing out the incorrect mention of a methoxy group in the manuscript text. We have implemented the suggested changes and corrected the error.

(10) p. 13, Figure 5a: A kinetic resolution reaction is shown for the secondary alcohols, in which the transesterification reaction converts the (S)-enantiomer of the alcohol to the ester and leaves the (R)-enantiomer untouched. However, no mention of enantioselectivity is made anywhere in the manuscript text or the SI, and no methods for the chromatographic separation of substrate and/or product enantiomers are reported in the analytical procedures (in fact, Figure S49 even fails to show achiral-phase separations of these compounds; see comment 27). It hence seems that the enantioselectivity in these reactions was not investigated at all. The authors should elucidate the enantioselectivity of PcAOx-VPN (and, ideally, also of ParAOx-VPN and AflAOx-VPN) in transesterification reactions of sec-alcohols, as this parameter is an important determinant of the application potential of these enzymes. In general, a broader range of substrates should be investigated, including esters with acid parts other than acetyl.

Following this reviewer (and reviewer 1) we have performed additional experiments clarifying the enantioselectivity of PcAOx-VPN. These have been included and discussed in the revised manuscript. We also have performed additional experiments targeting further nucleophiles (amines and thiols).

(11) p. 14, Figure 6: The panels a), b) and c) in the Figure should be explained in the Figure caption. Following the reviewer's suggestion, we simplified Figure 6 and also added explanatory text..0

(12) p. 15, lines 213–215, also Scheme 3: While the authors present conclusive experimental

evidence supporting the direct involvement of oxidised flavin and His561 in the transesterification reaction, this is still a weak basis for proposing a detailed catalytic mechanism. What's more, the arrow-pushing mechanism drawn in Scheme 3 suggests a direct nucleophilic displacement of the alcoholate leaving group from the ester. As both enzymatic and non-enzymatic transesterification reactions are known to proceed via tetrahedral intermediates, this mechanistic proposal is very unconventional. In a way, the authors seem to be aware of this, as they have used a tetrahedral intermediate as ligand for the docking simulations described on p. S41 of the Supporting Information (Figure S20). However, these docking results are not mentioned in the manuscript text in any way. If a tetrahedral intermediate is involved, its negative charge on oxygen needs to be stabilised in some way within the enzyme's active site (cf. the "oxyanion hole" of lipases). The authors should look out for residues able to stabilise a tetrahedral intermediate in the crystal structure of the enzyme and in the docking poses. They should also investigate whether the reaction is inhibited by hydrolase inhibitors of the phosphonate type, which mimic the tetrahedral intermediate. On the basis of the currently presented data alone, the mechanistic proposal is highly speculative.

We fully agree with the referee's assessment. Following this advice, we have revisited the crystal structure of PcAOx. The pivotal role of His561 and Asn604 (and the corresponding residues in other oxidases) in deprotonating the alcohol group during oxidation has been discussed previously (e.g. by Fraaije and co-workers). By analogy, we consider it likely that this His–Asn dyad also contributes to the transesterification reaction by facilitating the nucleophilic attack on the acyl donor carbonyl group.

Moreover, our docking studies with the tetrahedral intermediate suggest stabilising interactions: hydrogen bonding between the oxyanion and Asn604, together with electrostatic stabilisation provided by the positively charged His–Asn pair. These findings have led us to refine our mechanistic proposal, which is now presented as working hypothesis for further studies in the revised manuscript. We apologise for the overly simplistic mechanistic scheme in the initial version, which did not sufficiently reflect the established involvement of tetrahedral intermediates. In the revised manuscript, we now explicitly acknowledge the speculative nature of the proposed mechanism and emphasise that further mechanistic studies will be required to clarify the catalytic details.

(13) p. 15, line 223: "...and likely have a similar active site architecture." Why make this speculative statement here? The multiple-sequence alignment shown in Figure S8 (which is barely discussed in the manuscript text, and misnumbered as Figure S17) contains all the information needed to compare the active-site residues of the three enzymes. For instance, it shows residues W560, H561, and N604 to be fully conserved. Also, with such a high sequence similarity between the three enzymes, it would be easy to generate high-quality homology models of ParAOx and AflAOx to compare their active sites to that of PcAOx on a 3D-structural basis. Finally, the authors speak of "sequence similarity" in the text, but I assume the percentages given in parentheses refer to sequence identity. This should be made clear, e.g.: (ParAOx: 87% identity, AflAOx: 53% identity).

We thank the reviewer for this helpful suggestion. The structural comparison has been introduced following her/his suggestion.

(14) p. 16, line 228: "Figures S17, S18" I believe the authors want to refer to Figures S4 and S5 here. In general, misnumbering of Figures in cross-references is a common problem with this manuscript. Double-check all cross-references and revise.

Corrected as suggested by the reviewer.

(15) SI, p. S4: "...by adding 1 μ L Dpn I enzyme for every 1 μ g PCR product." Please specify the concentration and/or the specific activity of the DpnI enzyme solution used. "...containing 50 μ g \times mL⁻¹ at 37 °C..." Containing 50 μ g/mL of what? I assume kanamycin.

Thank you for spotting this. We cannot specify the amount of Dpn I as this information was not provided by the supplier. According to the manual 1 μ l corresponds to 1 unit, which is what we have added to the description. We apologise for omitting kanamycin and obviously have corrected this.

(16) SI, p. S5: "...oxidative coupling of phenol and 4-amino antipyrine..." No mention of phenol in the screening medium is made, instead 3,5-dichloro-2-hydroxybenzenesulfonic acid is mentioned as the coupling partner. Double-check and revise the text and Figure S1. "After 30 h cultivation at 30°C, single red colonies were selected for sequencing." Please provide more details on the resulting variant library: How many positives were sequenced? How many of them were wild types? What variants were found in the pool of sequenced positives?

The textual inaccuracies have been corrected: the correct coupling partner in the screening medium phenol.

With regard to the variant library, we would like to clarify our strategy. In line with our objectives, we did not sequence or characterise multiple monoclonal colonies from the plates. Instead, only those monoclonal colonies that exhibited the expected phenotype were selected for further analysis. This approach is consistent with the primary purpose of the method, namely to provide a straightforward means of constructing and identifying mutants, rather than to perform an exhaustive characterisation of all clones. The mutant described here was discovered because its colony displayed a more intense red phenotype, and its catalytic properties were subsequently investigated in detail. This was thus an incidental finding arising during characterisation, rather than the result of plate screening specifically aimed at identifying transacylase activity.

(17) SI, p. S6: "Crude cell extracts were prepared..." Please provide further experimental details of the cell disruption and purification procedure: How much buffer was used for what quantity of cell pellet? What types of Ni column and desalting column were used (dimensions, manufacturer; carrier material for the Ni column: I assume Ni-NTA)? Was elution performed with a linear gradient over the specified imidazole concentration range or was a step-elution protocol used? How many column volumes of eluent were used for elution? Was the chromatography performed manually (e.g., with a syringe or a peristaltic pump) or using an FPLC instrument?

Corrected as requested by the reviewer.

(18) SI, pp. S6–S8: Please use precise and consistent terminology to refer to the bands in the SDS-PAGE images. "the total bacteria" and "whole cell protein" are not precise, nor are they consistent with the terminology used in Figure S2, where "cell extract of E. coli overexpressing PcAOx" is used. Also, "250 mM imidazole eluting pure enzyme" in the caption of Figure S5 is a poor description, as this protein fraction is clearly not pure, with many additional bands visible.

Corrected as requested by the reviewer.

(19) SI, p. S8: "...was further purified using gel filtration chromatography..." Please provide experimental details: What type and dimension of column were used? What was the mobile phase? What was the volume and concentration of applied (crude) protein solution? What was the volume of the collected fractions?

Added as requested by the reviewer.

(20) SI, p. S12: "The concentrations of reagents and enzymes as well as reaction conditions are specified in the figure captions." Please provide further experimental details: I assume that stock solutions were used to prepare the reaction mixtures. What were their concentrations and in what order were they mixed to prepare the reactions? "All quantifications are based on calibration curves using authentic standards." Was the quantification performed relative to an internal standard or based directly on the analyte integrals?

Added as requested by the reviewer.

(21) SI, p. S13, Figure S8: To make this figure more informative, the residues that were mutated to generate transesterification activity and other active-site residues should be highlighted.

Added as requested by the reviewer.

(22) SI, p. S34, Figure S12: How was the residual activity determined (type of assay, reaction time)? "10% (w/v) PMFS" The abbreviation is PMSF, not PMFS. $10\% \text{ w/v} = 100 \text{ g/L} = 0.574 \text{ mol/L}$. This concentration is too high to be plausible.

Corrected as requested by the reviewer.

(23) SI, p. S35, caption of Figure S13: The dot markers for benzyl acetate and the dashed line for the sum of ester and aldehyde should be shown in red colour, as they are in the Figure itself.

Corrected as suggested by the referee.

(24) SI, p. S38, Figure S17: The figure misses a legend that explains what the three differently coloured bars refer to.

Corrected as suggested by the referee.

(25) SI, pp. S39–S40, captions of Figures S18–S19: How were the temperature tolerance and pH

tolerance of the enzymes determined? By pre-incubation of the enzyme solution at the indicated temperature (and pH) for the indicated time? How was activity determined (type of assay, reaction time)?

Corrected as suggested by the referee.

(26) SI, p. S41: Please provide information on how the enzyme structures were prepared for docking (e.g., adjustment of protonation state, calculation of charges, setup of simulation cell).

Added as requested by the reviewer.

(27) SI, pp. S43–S76, gas chromatograms: Chromatograms of reactions using high substrate concentrations show heavy fronting, a sign of exceeding the capacity of the GC column. The quantifications based on these chromatograms are likely imprecise. The samples should have been diluted before analysis. Also, the cross-references to the Figures in the manuscript are incorrect: For instance, "Chromatograms related to Figure 2" seem to refer to a Figure that is no longer part of the manuscript, "Chromatograms related to Figure 3" actually refers to Figure 2, etc. Double-check and revise. The chromatograms shown on pp. S55 and S56 cannot possibly correspond to the given reactant concentrations ([vinyl acetate] = 50 mM, [2-phenylethanol] = 250 mM), as the final two time points show complete consumption of the alcohol, which is not possible with sub-stoichiometric amounts of acyl donor. "Chromatograms related to Figure 4" again do not refer to the correct figure. Also, why is there heavy tailing now for a chromatographic separation that gave good peak shape in earlier chromatograms? In Figure S49 (pp. S66–S67), only primary alcohols (and unbranched esters) are shown, in contrast to what the figure caption claims.

We apologise for the confusion caused by the chromatogram documentation — apparently, we had been confused ourselves. In view of the reviewer's helpful observations and considering that a complete presentation of all individual measurements proved overly complex and potentially confusing, we have decided to restrict the Supporting Information to a set of representative examples. We believe this revision improves the overall clarity and readability of the data.

(28) Typos: p. 5, line 68: alcohol → alcohols; p. 6, Figure 1 caption: yellow → white, W590 → W560; p. 7, line 114: effect the → effect of the; p. 7, Figure 2: 1-phenylethanol acetate → 2-phenylethyl acetate, 1-phenyl ethanol → 2-phenylethanol; p. 9, line 130: doner → donor; p. 11, line 160: upon completion of the activate acyl donor → upon complete consumption of the activated acyl donor; p. 12, line 183: these → this; p. 12, line 185: in all cases → in both cases; p. 14, line 204: catalysts → catalyst; p. 15, line 217: His601 → His561; p. 15, line 218: it → its; SI, p. S12: Biotransformations → Biotransformations; SI, p. S12: cures → curves; SI, p. S31: compared and → and; SI, p. S36: 2-phenyl ethanol → benzyl alcohol; SI, p. S67: china → chain

Corrected.

Once again, we thank the reviewers for their efforts and for their excellent comments, which have helped us to improve the manuscript. Their contributions are greatly appreciated. A detailed response to their comments is provided below:

Reviewer #1 (Remarks to the Author):	
Dear Authors,	
In general I am satisfied with the revision; however, I have a few additional minor remarks as stated below:	
(1) Please erase the comment on page S12 in Supporting Information.	We apologise for this omission. This section has been corrected.
(2) The catalytic channel/tunnel depicted in Figure S1 could be better visualized (e.g., a larger picture is needed in general, including the measured diameter of the channel at the entrance and in the bottlenecks, as well as a more in-depth discussion on the pronounced hydrophobic amino acid characteristics).	Following the reviewer's advise, we have improved to clarity of Figure S7 and also extended the manuscript section dealing with the active site hydrophobicity.
(3) Please add the discussion to Figure S2, as it is unclear to me. What does the middle line mean, except as a timescale for the reaction? Is it the interface of the external reaction medium and the enzyme active site? One can guess what all these mean, but it should be explicitly specified/clarified.	We assume the referee is referring to Figure S16. The Cleland diagram of the proposed, sequential reaction mechanism has now been discussed in more detail.
(4) The authors investigated the enantioselectivity of the PcAOx-VPN-catalyzed esterification of 1-phenylethanol and lactic acid using both enantiomers separately and as racemates; however, they did not present the results for E-values (or I missed something?). Please comment on this.	We apologise for the omission of the estimated E-values. They apparently got lost in between manuscript versions. The E-values have now been added to the revised manuscript.
(5) The authors show PcAOx crystal structures for wild-type PcAOx and engineered variant PcAOx-VPN in Figure S3, and speak about some angles "a and b" between these two enzymes and the W560 residue. I was slightly puzzled when I saw no angles and their values. Please specify (visualized) with dashed lines which of these angles you are speaking about, and calculate their values if it is so important. Otherwise, it is too speculative, and it adds absolutely nothing to this research.	In this context, the term "angle" was intended to describe the different viewing perspectives used in the images to present the stereoviews of W560 and the FAD ligand in the PcAOx crystal structures, rather than actual

	geometric angles. To prevent any ambiguity, we have removed the corresponding annotations in the revised SI and corrected the image order.
(6) The experimental procedure describing the preparative-scale reaction presented in Section 2.12 of the Supporting Information is concise and should be revised to include more details to ensure full reproducibility for other researchers. For example, what does it mean that the product was distilled? Please specify the conditions in terms of the pressure and temperature. I also see no data concerning the isolated yield that is given in the main manuscript, but why is it not included in the SI document?	Following the reviewer's suggestion, we have further added details to the revised SI.
(7) In the previous round of revision, I asked authors the following: "(...) In the 'Supporting Information' file, I recommend compiling all relevant details regarding the recombinantly produced enzymes into a single table. This table should include information on the vector and expression host used, plasmid number, plasmid vector, and the inducer used for protein expression. (...)". To my great disappointment, I do not see the table with the collected information provided in SI. Please complete this.	We have now added this table.
(8) Figure 4 in the main manuscript contains blurred chemical structures. I recommend avoiding the use of coloring compounds on the scheme in this style, especially when using red and blue highlights.	Changed as suggested by the reviewer.
Reviewer #2 (Remarks to the Author):	
The authors of this work presented the discovery of a mutant of alcohol oxidase from Phanerochaete chrysosporium (PcAOx-VPN) whose more hydrophobic cavity is instrumental to avoid hydrolytic side activity in transesterification reactions. As stated in the first round of review, I think that this is a well-done and relevant piece of work. All issues were addressed by authors.	
Reviewer #3 (Remarks to the Author):	
With their revised manuscript, Ma, Hollmann, Wang, and co-workers show an honest and substantial effort to improve their paper based on the criticism raised by myself and the other reviewers of the original submission. The authors address every reviewer comment openly and adequately, they report a substantial body of additional experimental work, which clarifies many of the questions that the original manuscript left unanswered and thereby significantly improves the overall quality of the paper, and they have incorporated numerous small changes to the text and figures to raise the overall level of clarity and data presentation. Most importantly, the authors now demonstrate a significantly broader substrate scope than in	

their original submission, including non-alcohol nucleophiles and a more diverse set of acyl donors, and they provide convincing experimental evidence for (S)-stereoselectivity with chiral alcohol substrates. The mechanistic proposal has also been amended, and it is now communicated as a working hypothesis for more detailed enzymological studies to be carried out in the future.

With these significant improvements, I can now recommend this work for publication in Nature Communications. A few minor points still require correction, but I am optimistic that these can be handled without further peer review. Suggestions for final improvements and corrections are provided below:

(1) Abstract: Some of the minor text changes made during revision seem to have altered or obscured the originally intended meaning, and a few passages have been unclear even in the original: "but they are less elegant" – less elegant compared to what alternatives? // "In this study, we present a mutant of alcohol oxidase from Phanerochaete chrysosporium (PcAOx-VPN) that exhibits no hydrolytic activity." – At this point in the abstract, the (unexpected) transesterification activity of PcAOx-VPN has not yet been introduced, so the statement is lacking context. // "...enabling transesterification reactions in water, previously considered impossible, ..." – Transesterification in water has been demonstrated to be possible, even if it has been plagued by competing hydrolysis. Please revise these passages for improved clarity.	Thank you for the careful check! We agree that 'less elegant' is very subjective and has been omitted. Also we agree that the transesterification activity should be introduced first, which we have now done.
(2) Scheme 1 caption: "Comparison of the time courses of classical transesterification time courses..." I think the second instance of "time courses" should be deleted.	Corrected as suggested by the reviewer.
(3) Non-standard nomenclature: p. 6, line 98: 1,2-propanol → propane-1,2-diol, line 100: propane diol → propanediol; p. 14, line 225: 1-phenyl ethanol → 1-phenylethanol	Corrected as suggested by the reviewer.
(4) p. 9, line 149: "As enzyme inactivation..." This sentence seems to lack a verb.	Corrected as suggested by the reviewer.
(5) p. 13, line 199: "This is consistent with Fraaije's observation that..." Because of text insertions, the cross-reference using "this" has become ambiguous. Better re-phrase, e.g. "The absence of hydrolysis activity is consistent..."	Corrected as suggested by the reviewer.
(6) p. 13, line 207: "Also benzyl acetate and 2-phenylethyl acetate were accepted as acyl donors (Figure 5, [b] and [c])." Since the authors discuss restrictions in active-site access for bulky substrates just a few lines above, it would help to clarify that benzyl acetate and phenethyl acetate are accepted as acyl donors only when combined with small, aliphatic alcohols as substrates.	Added as suggested by the reviewer.

(7) p. 13, line 211: "Amongst the aliphatic alcohols tested, primary alcohols (4-8) were preferred..." In Figure 5, substance numbers 4–8 refer to the ester products, not the alcohol substrates. Although it is clear what the authors want to say, I would consider it advantageous to develop a clear and systematic abbreviation scheme that allows alcohols, acyl donors, and products to be unambiguously referred to, e.g., by using numbers for alcohols, letters for acyl donors, and a combination of number and letter for the resulting ester products.	We thank the reviewer for this observation. We hope, also the reviewer will find the new numbering system clear and logical.
(8) p. 18, lines 277–287, also Scheme 2 and Figure S8: What the authors call a "transition state" is really an intermediate and should be referred to accordingly. [Note that a transition state represents a state of maximum energy in a reaction trajectory, where bonds are partly formed and broken. (cf. the IUPAC definition of 'transition state': https://goldbook.iupac.org/terms/view/T06468) In contrast, the tetrahedral intermediate shown in Scheme 2 and referred to in the text is a chemical entity that represents a local energy minimum and comprises only fully formed bonds, hence why it is called an 'intermediate' (https://goldbook.iupac.org/terms/view/I03096).] As intermediates are not usually drawn in brackets, I suggest to remove the brackets from Scheme 2b. Also, the arrow pushing in Scheme 2b is only partially correct, as not all bonding changes occurring between the shown states are accounted for by curly arrows (for instance, the "flipping" of the double bond in the histidine's imidazole ring is not accounted for). Please formulate a complete arrow-pushing mechanism. Besides, in Figure S8, all four structures miss an oxygen atom (the one of the alcohol part of the acyl donor).	We apologise for the floppiness in our chemical nomenclature and mechanism drawing, which we obviously corrected as suggested by the reviewer.
(9) SI, Section 1.6: Please provide instrumentation and conditions for the chiral analyses that have been carried out during revision. Please also provide exemplary chromatograms of enantiomeric separations (similar to those shown in Section 2.10) in a format that clearly demonstrates separation of a racemic standard under the same conditions used for separation of the biotransformation samples. If any of the samples were derivatised before analysis (I consider this likely for the carboxylic acid derivatives 28 and 29), please provide the derivatisation procedure that was used.	We have added the instrumentation, conditions, along with an example chromatogram for the chiral analyses to the corresponding section.
(10) SI, Section 1.7: This section now contains a description of a molecular docking workflow that is even less informative than the corresponding paragraph in the original submission. On the other hand, a paragraph on docking that provides adequate details of the workflow is provided in Section 2.2. Please combine the two paragraphs into one and clarify which	Corrected as suggested by the reviewer. Section 1.7 has been deleted and the information is now included in the respective sections 2.1 and 2.2

of the two workflows was used for what docking problem. Also, the SI Word file still contains author comments in Section 1.7 that the authors might want to remove before publication.	
(11) Typos: p. 11, line 190: "may not sufficiently spacious" → "may not be sufficiently spacious", p. 20, line 317: "We therefore hypothesis" → "We therefore hypothesise"	Thank you once more for you very careful check and contribution to the manuscript.

The submitted manuscript, entitled "Irreversible transesterification reactions in aqueous media" (NCOMMS-24-71991-T), authored by F. Hollmann, Y. Wang, et al., presents the discovery of a novel engineered variant of alcohol oxidase from *Phanerochaete chrysosporium* (PcAOx-VPN) exhibiting unexpected ('promiscuous') quasi-irreversible transesterification activity in water, without reversed hydrolytic action. The reported biocatalyst is the first known flavin-containing enzyme capable of catalyzing a reaction beyond its native catalytic repertoire. As such, it has the potential to serve as a superior alternative to existing variants of *Mycobacterium smegmatis* acyltransferase (MsAcT).

I find this paper highly interesting and valuable to the broader scientific community, as the reported research presents transformative results that significantly advance the field of bioorganic chemistry. Moreover, it has the potential to open new avenues for future synthetic endeavors, particularly in the transesterification of alcohols. In this context, the study exemplifies the critical need for discovering novel biocatalysts with unnatural features that enable challenging transformations—often unattainable through classical chemical methodologies due to strongly unfavorable thermodynamics. A key strength of this paper is the successful development of the efficient triple mutant M59V-Q60P-R61N (PcAOx-VPN) through saturation mutagenesis. Notably, this variant demonstrated the ability to catalyze the desired transesterification of selected model primary alcohols with bulkier aryl substituents, such as benzyl alcohol and 2-phenylethanol. However, while the newfound activity of the M59V-Q60P-R61N variant is highly promising, there are notable limitations in the tested substrate scope. The study suggests that the method is restricted to a narrow range of substrates, predominantly specific primary alcohols. Interestingly, PcAOx-VPN also catalyzes the efficient alcoholysis of benzyl esters using various aliphatic alcohols (MeOH, EtOH, *n*-PrOH, and *n*-BuOH), which could serve as a promising alternative to hydrolytic processes used in applications such as biodiesel production.

The experimental and analytical methods were rigorously designed and executed to meet high standards, ensuring reliability and reproducibility. A thorough analysis of the "Supporting Information" file confirms that the results are well-supported by appropriately designed controls, detailed analyses, and robust statistical evaluation, with no significant flaws or oversights. Technically, the reported data are well-documented, and the paper is written in a clear, concise, and accessible manner. Furthermore, all included figures, tables, and supplementary materials are of high quality, well-organized, and effectively communicate the results.

Given its exceptional quality, novelty, and significant impact, I recommend that this manuscript be considered for publication in *Nature Communications*. However, I suggest that the following issues be carefully addressed by the authors prior to publication:

Minor points:

- (1) In general, the title of the submitted manuscript is slightly misleading, and more suitable for review-type article rather than a communication in Nature journal. In general, it suggests as it report on this topic in a broader context. In my opinion, the title should relate more to discovered variant of alcohol oxidase from *Phanerochaete chrysosporium* (PcAOx-VPN) and its promiscuous catalytical activity in terms of transesterification in water.
- (2) In the 'Introduction' section authors stated that: "(...) Serendipitously, we discovered transesterification activity in variants of the alcohol oxidase from *Phanerochaete chrysosporium* (PcAOx)¹¹⁻¹³ (...)". After careful reading of all three cited articles, I cannot confirm that any of the authors from the list of the submitted paper participated in those discoveries. Therefore, please rephrase this sentence as it suggests that the authors took part in those studies. As I understand correctly, Prof. Marco W. Fraaije (who is one of the authors of the papers cited in Refs. 11–13) is affiliated with the University of Groningen (NL) and not with TU Delft (NL).
- (3) In the 'Introduction' section, there is an insufficient number of references concerning the application of *Mycobacterium smegmatis* acyltransferase (MsAcT). I strongly recommend adding some key citations that highlight the potential of this enzyme in synthetic applications, such as: (i) P. Borowiecki et al. *Angew. Chem. Int. Ed.* **2025**, *64*, e202420133; (ii) U. T. Bornscheuer et al. *ACS Catal.* **2021**, *11*, 14906–14915; (iii) U. T. Bornscheuer et al. *Angew. Chem. Int. Ed.* **2020**, *59*, 11607–11612; (viii) W. Kroutil and F. Himo et al. *ACS Catal.* **2018**, *8*, 10698–10706; (ix) U. Hanefeld et al. *Adv. Synth. Catal.* **2017**, *360*, 242–249, etc.
- (4) Please refine the following sentences: "(...) The incubation of for example 2-phenylethanol acetate with PcAOx-VPN (...)". The chemical name of "2-phenylethanol acetate" is faulty in this case. It should be "2-phenylethyl acetate".
- (5) Out of curiosity, I would like to ask the authors whether they have tested other quasi-irreversible acyl donors (e.g., fatty acid-like vinyl esters, isopropenyl acetate, acetic anhydride, 4-nitrophenyl acetate, 4-chlorophenyl acetate, 2,2,2-trifluoroethyl acetate, etc.).
- (6) The authors stated that: "(...) For the acyl donor vinyl acetate, however, substrate inhibition was detected (...)" and then: "(...) We interpret these observations as a result of an ordered, sequential reaction mechanism (...)". Maybe the drop in the catalytic activity is due to the *in situ* formation of acetaldehyde. As this might be true, once again, I suggest testing other acyl donors which does not generate acetaldehyde byproduct.
- (7) The authors present the results of biotransformations using concentrations of the product obtained. I think it would be better to show a conversion (% conv.).

Major points:

- (1) This study is significantly limited by its narrow substrate scope, as it primarily focuses on ortho-, meta-, and para-substituted benzyl alcohols. In particular, the absence of racemic secondary alcohols is a major drawback, as their inclusion could provide valuable insights into the catalytic capabilities of the engineered variant, especially regarding enantioselectivity. To address this limitation, I strongly recommend revising the manuscript by applying

the *PcAOx*-VPN variant to the kinetic resolution of representative racemic secondary alcohols. These should include benzylic alcohols, aliphatic alcohols, "small-bulky" aliphatic-aryl alcohols, hydroxy esters, heteroaromatic (halo)methyl alcohols, and "bulky-bulky" aryl-aryl alcohols. Furthermore, all reactions and the resulting optically active products should be fully characterized in terms of their stereochemical outcomes. This should include an assessment of the enzyme's stereopreference, the enantiomeric excess (% ee) of the isolated kinetic resolution (KR) products, and the enantioselectivity (E-value).

- (2) I also strongly encourage the authors to test their enzyme on substrates containing more than a single hydroxyl group. Using exemplary diols, triols (such as simple glycerol), and other benchmark multi-hydroxyl compounds could help demonstrate the regioselectivity of the investigated biocatalyst. Additionally, the inclusion of at least one amino alcohol as a substrate would be valuable in assessing the chemoselectivity of the discovered biocatalyst.
- (3) The authors stated that the unusual ability of *PcAOx*-VPN to catalyze transesterification in water without hydrolysis is attributed to the hydrophobic architecture of its active site. To support this hypothesis regarding the catalytic mechanism of action, I strongly recommend visualizing the tunnels and channels in the molecular structure of *PcAOx*-VPN that lead to the active site entrance. This can be achieved using software such as CAVER Analyst 2.0, developed by researchers at the Loschmidt Laboratories and the National Centre for Biomolecular Research (NCBR) at Masaryk University. Such an analysis could provide deeper insights into the structural factors and molecular basis that determine the rate-limiting step, particularly concerning diffusion effects. Additionally, it would enhance the rationalization of the observed substrate scope.
- (4) Since *Nature Communications* values research that bridges disciplines and has applications across multiple fields, I recommend that the authors enhance their paper to align with these standards, ensuring that their work resonates with a broader scientific audience beyond specialists in biocatalysis. To achieve this, I suggest highlighting potential applications of the transesterification methodology in a broader context. For instance, demonstrating its extension toward the synthesis of exemplary small-molecule pharmaceuticals, bioactive compounds, or specialty chemicals would significantly strengthen its relevance. Additionally, this would help justify the selection of model alcohols. Without such examples, the impact of this methodology may be perceived as limited to a narrowly defined set of substrates, potentially restricting its broader applicability and diminishing its potential influence on future discoveries in modern bioorganic chemistry.
- (5) In the "Supporting Information" file, I recommend compiling all relevant details regarding the recombinantly produced enzymes into a single table. This table should include information on the vector and expression host used, plasmid number, plasmid vector, and the inducer used for protein expression.
- (6) According to the Notice for Authors set by prestigious journals: "(...) For synthetic methods, a selection of substrates illustrative of the scope of the reaction should be made, and the isolated yields of the corresponding purified products should be reported. Product yields determined by HPLC or GC are considered indicative of a preliminary catalytic study (...)." Therefore, to better illustrate the applicability of the method, I strongly recommend including at least one reaction performed on a gram scale.